

# FESDIA (v1.0): Exploring temporal variations of sediment biogeochemistry under the influence of flood events using numerical modelling

Stanley I. Nmor[1], Eric Viollier[1], Lucie Pastor[2] Bruno Lansard[1] Christophe Rabouille[1] Karline Soetaert[3]

[1] Laboratoire des Sciences du Climat et de l'Environnement, LSCE/IPSL,CEA-CNRS-UVSQ-Université Paris Saclay, 91198 Gif sur Yvette, France
[2] Laboratoire Environnement Profond, Ifremer – Centre de Bretagne, 29280 Plouzané, France.
[3] Royal Netherlands Institute of Sea Research (NIOZ), Department of Estuarine and Delta Systems, Korringaweg 7, P.O. Box 140, 4401 NT Yerseke, The Netherlands

*Correspondence to*: Stanley I. Nmor (**stanley.nmor@lsce.ipsl.fr**)

**Abstract.** Episodic events of flood deposit in coastal environments are characterized by deposition of large quantities of sediment containing reactive organic matter within short periods of time. While steady-state modelling is common in sediment biogeochemical modelling, the inclusion of these events in current early diagenesis models has yet to be demonstrated. We adapted an existing model of early diagenetic processes to include the ability to mimic an immediate organic carbon deposition. The new model version was able to reproduce the basic trends from field sediment porewater data affected by the November 2008 flood event in the Rhone River prodelta. Simulation experiments on two end-member scenarios of sediment characteristics dictated by field observation, (1-high thickness deposit, with low TOC and 2-low thickness, with high TOC), reveal contrasting evolutions of post-depositional profiles. A first-order approximation of the differences between subsequent profiles was used to characterize the timing of recovery (i.e relaxation time) from this alteration. Our results indicate a longer relaxation time of approximately 4 months for $SO_4^{2-}$ and 5 months for DIC in the first scenario and less than 3 months for the second scenario which agreed with timescale observed in the field. A sensitivity analysis across a spectrum of these end-member cases for the organic carbon content (described as the enrichment factor $\alpha$) and for sediment thickness - indicates that the relaxation time for oxygen, sulfate, and DIC decreases with increasing organic enrichment for a sediment deposition that is less 5 cm. However, for larger deposits (> 14 cm), the relaxation time for oxygen, sulfate and DIC increases with $\alpha$. This can be related to the depth dependent availability of oxidant and the diffusion of species. This study emphasizes the significance of these sediment characteristics in determining the sediment's short-term response in the presence of an episodic event. Furthermore, the model described here provides a useful tool to better understand the magnitude and dynamics of flooding event on biogeochemical reactions on the seafloor.

## *1* Introduction

Coastal margins play a crucial role in the global marine systems in terms of carbon and nutrient cycling (Wollast, 1993; Rabouille et al., 2001b; Cai, 2011; Regnier et al., 2013; Bauer et al., 2013; Gruber, 2015). Due to their relatively shallow depth,



sedimentary early diagenetic processes are critical for the recycling of a variety of biogeochemical elements, which are influenced by organic matter (OM) inputs, particularly carbon (Middelburg et al., 1993; Arndt et al., 2013). Furthermore, these processes have the potential to contribute to the nutrient source that fuels the primary productivity of the marine system. In river-dominated ocean margins (RiOmar, McKee et al. (2004)), organic matter input can also be enhanced by flood events which provide a significant fraction of the particulate carbon (POC) delivered to depocenters (Antonelli et al., 2008). The fate of organic matter derived from riverine input to the sediment is of biogeochemical importance in coastal marine systems (Cai, 2011) because it serves as a sink for particulate organic carbon and nutrients as well as an intense site of carbon and nutrient remineralization (Burdige, 2005; McKee et al., 2004; Sundby, 2006).

In the context of early diagenetic modelling, numerical models with time-dependent capability are well established (Lasaga and Holland, 1976; Burdige and Gieskes, 1983; Rabouille and Gaillard, 1991; Boudreau, 1996; Soetaert et al., 1996; Rabouille et al., 2001a; Archer et al., 2002; Couture et al., 2010; Yakushev et al., 2017), and they are used in many coastal and deep-sea studies. However, because of the scarcity of observations and their unpredictability, the role of massive episodic events in these models has frequently been overlooked (Tesi et al., 2012). As these rare extreme events are being currently documented in various locations, there is a growing appreciation for their impact on the coastal margin (Deflandre et al., 2002; Cathalot et al., 2010; Tesi et al., 2012).

Attempts to use mathematical models to understand perturbation-induced events on early diagenetic processes have resulted in a variety of approaches that incorporate this type of local phenomenon. As an example, previous research in deep-sea systems suggests that megafaunal perturbation can cause a 35% increase in silicic flux when compared to steady-state estimates (Rabouille and Gaillard, 1990). Katsev et al. (2006) demonstrated that the position of the redox boundary in organic-poor marine sediment can undergo massive shifts due to the flux of new organic matter on a seasonal basis, whereas on a longer time scale (e.g. decadal), redox fluctuation linked to organic matter deposition can induce the redistribution of solid-phase manganese with multiple peaks. Another study in a coastal system revealed that coastal sediments change as a result of an anthropogenic perturbation in the context of bottom dredging and trawling (Van de Velde et al., 2018). More recently, using similar model, De Borger et al. (2021) highlighted that perturbation events such as trawling can possibly decrease total OM mineralization.

In river-dominated ocean margins, episodic flood events can deliver sediment with varying characteristics depending on its source origin, frequency and intensity (Cathalot et al., 2013). Therefore, the flood characteristics have direct impact on the deposited sediment's characteristics such as scale/thickness of the deposited layer, composition (mineralogy and grain-size), OM content and so on. For example, In the Rhone prodelta, a single flash flood can deliver up to 30 cm of new sediment material in a matter of days (Cathalot et al., 2010; Pastor et al., 2018). Despite the large amount of sediment introduced by this episodic loading, porewater species like oxygen ($O_2$) can be restored after a few months (Cathalot et al., 2010). It has also been noticed (Rassmann et al, 2020) that spring and summer porewater compositions measured for several years following fall and winter floods show quasi-steady state profiles for sulfate and DIC. Similar massive deposition was also reported in the





Saguenay Fjord (Quebec, Canada) (Deflandre et al., 2002; Mucci and Edenborn, 1992). This recovery timescale from this perturbation has only been roughly estimated for short-lived species like oxygen, but this is not always the case for sulfate ($SO_4^{2-}$), dissolved inorganic carbon (DIC), or other redox species. Furthermore, due to the limitation in temporal resolution of the observations, the short-term post depositional dynamics in the aftermath of this flood deposition event are scarcely described, making it difficult to discern how the system responds after the event. While experimental approaches (Chaillou et

al., 2007) can provide useful insight into how they work, they lack the ability to provide continuous system dynamics and are often difficult to set up. A modelling approach can assist in addressing these issues, providing useful feedback in terms of the scale and response of the sediment to this type of event.

The goal of this study is to better understand the dynamic interplay of episodic events in terms of their impact on benthic fluxes, post-flood evolution following deposition, and relaxation timescale. As the relaxation dynamics represent a gap in our

understanding of how coastal systems respond to external drivers, we characterize the timescale of the recovery of sediment porewater profiles using a first order approximation. Our numerical model represents the important early diagenesis processes in the sediment and can explicitly simulate non-steady state early diagenesis processes in systems subject to events such as massive flood or storm deposition. We demonstrate the model's utility in describing data collected from a flood event in November/December 2008 (Pastor et al., 2018) as well as numerically investigate the impact of varying degrees of flood type

characteristics on the system's relaxation dynamics. This work is a foundation for a more in-depth investigation of the model-data biogeochemistry of the porewater and solid phase components of core samples from Pastor et al. (2018), and it provides a useful baseline for understanding the spatiotemporal dynamics of coastal marine systems subject to event-driven organic matter pulses.

## 2    Materials and methods

### 2.1    Site and events description

The Rhone prodelta serves as a case study for the development of the model used to evaluate sediment perturbation dynamics. This particular coastal area acts as the transitory zone between the inland river channel and the continental shelf (Gulf of Lion) of the Mediterranean Sea. The Rhone River with a drainage basin of 97800 $km^2$ and mean water discharge of 1700 $\mathrm{m^3 s^{-1}}$ delivers up to $1.6 \times 10^{10}$ moles C of particulate carbon (POC) annually (Sempéré et al., 2000) to the pro-deltaic part (i.e where

the river meets the sea). The Rhone prodelta covers an area of approximately 65 $km^2$ with depth ranging from 2 to 60 m (Lansard et al., 2009) and is characterized by high sedimentation rates reaching up to 41 $\mathrm{cm\,yr^{-1}}$ in the proximal zone (Rasmann et al., 2016; Lat - 43° 18.680' N, Long - 4° 51.038' E and average depth of 21 m) (Radakovitch et al., 1999; Miralles et al., 2005). Typically, the organic matter delivered reflects the Rhone River inputs (Lansard et al., 2008; Cathalot et al., 2013) while the magnitude of material transported, and the quantity of organic carbon transferred laterally varies according to seasons

and period of massive instantaneous deposition.

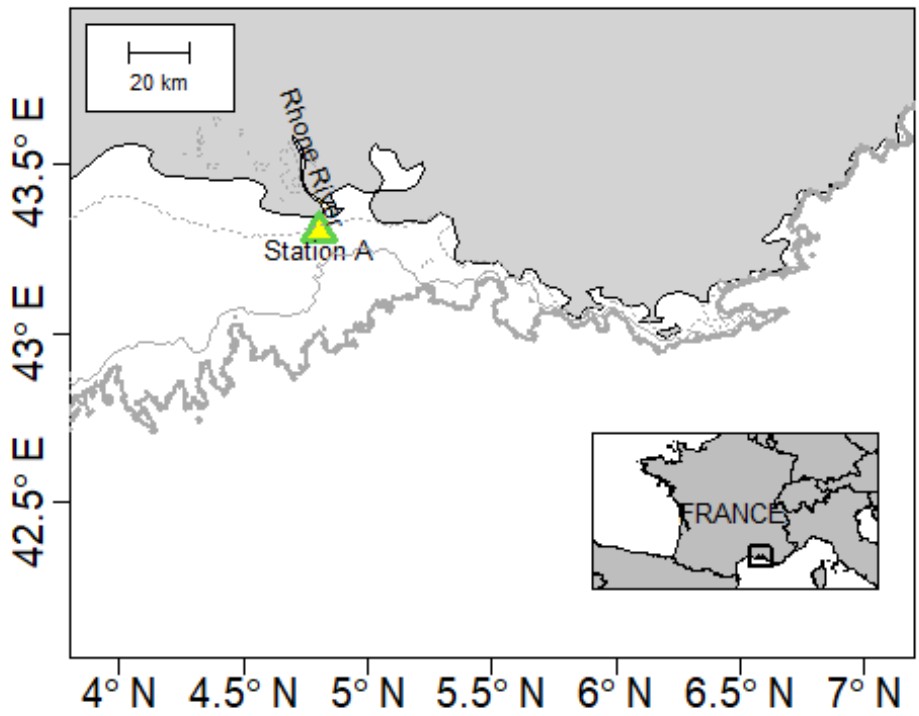

*Figure 1: Map showing the locations of sampling sites off the Rhone River mouth.*

Relating to the episodic pulse of organic matter, numerous studies have documented instances of flood driven deposition from the Rhone River from a hydrographic perspective (Boudet et al., 2017; Hensel et al., 1998; Pont et al., 2017). Pastor et al.

(2018) go beyond sedimentology and hydrographic characteristics to provide a concise description of the various flood types, their diagenetic signatures, and biogeochemical implications. Furthermore, published porewater chemistry and solid phase data have highlighted sediment characteristics following such an event (Cathalot et al., 2010; Toussaint et al., 2013; Cathalot et al., 2013; Pastor et al., 2018).

### *2.2*    **Model development and implementation**

Following the description of the Rhone River flood types and the composition of the flood deposit (mainly in terms of organic carbon) at the proximal station A (Pastor et al., 2018), we proceed to describe the model developed to explore the observed data and their diagenetic implications in terms of relaxation times and mode of behavior following this transient perturbation.

Our model combines the development in the OMEXDIA model (Soetaert et al., 1996), applied in the Rhone prodelta area (Ait Ballagh et al., 2021; Pastor et al., 2011) and which has recently been equipped with event-driven processes (De Borger et al.,

2021). In De Borger et al., (2021), the authors specifically addressed the issue of bottom trawling as a mixing and an erosional process that removes an upper layer of sediment and mixes a certain layer below. In addition, the model considers a bulk





categorization of reduced substance in a single state variable, ODU (oxidative oxygen unit). For our approach, the event is defined by an addition of a new layer on top of the former sediment-water interface. Furthermore, we explicitly modelled pathways involving Sulfur and Iron. Following this preamble, the following sections go over aspects of the model description 115 and parameterization.

### 2.2.1 Model state variables

The complete model describes the concentration of labile ($C_{org}^{fast}$) and semi-labile ($C_{org}^{slow}$) decaying organic matter, oxygen ($O_2$), nitrate ($NO_3^-$), and ammonium ($NH_4^+$), dissolved inorganic carbon (DIC), following the classic early diagenetic equation of (Berner, 1980; Boudreau, 1997). In addition to the model from De Borger et al. (2021), our model includes sulfate ($SO_4^{2-}$), 120 hydrogen sulfide ($H_2S$) and methane ($CH_4$), as well as iron species ($Fe^{2+}$ and $Fe(OH)_3$) (Table. 1).

Table 1: State variables described in the model.

| State variable | Model notation | Units | Description |
|---|---|---|---|
| **Solid** | | | |
| $C_{org}^{fast}$ | FDET | mmol C m$^{-3}$ | Fast decaying detritus |
| $C_{org}^{slow}$ | SDET | mmol C m$^{-3}$ | Slow decaying detritus |
| $Fe(OH)_3{}^{\neq}$ | FEOOH | mmol Fe m$^{-3}$ | Oxidized Ferric Iron |
| **Liquid** | | | |
| $O_2$ | O2 | mmol O$_2$ m$^{-3}$ | Oxygen |
| $NO_3^-$ | NO3 | mmol N m$^{-3}$ | Nitrate |
| $NH_4^+$ | NH3 | mmol N m$^{-3}$ | Ammonium |
| $SO_4^{2-}$ | SO4 | mmol S m$^{-3}$ | Sulfate |
| $H_2S$ | H2S | mmol S m$^{-3}$ | Hydrogen sulfide |
| $Fe^{2+}$ | Fe | mmol Fe m$^{-3}$ | Reduced Ferrous Iron |
| DIC | DIC | mmol N m$^{-3}$ | Dissolved inorganic carbon |
| $CH_4$ | CH4 | mmol N m$^{-3}$ | Methane |

In some coastal settings, oxidation via sulfate reduction has been highlighted as the primary pathway for organic carbon (OC) mineralization, with minor contributions from manganese and iron oxidation (Burdige and Komada, 2011). In addition, the





flux of integrated remineralization products such as DIC has previously been estimated to contribute up to 8 times that of diffusive oxygen uptake (Rassmann et al., 2020) - thus highlighting its importance in describing the amplitude of benthic recycling in coastal water. As such in this paper, we focus our analysis on these proxy variables ( $O_2$, $SO_4^{2-}$, DIC) because they serve as indicators of the integrated effect of the main diagenetic processes.

### *2.2.2*  **Biogeochemical reaction**

Early diagenesis processes on the seafloor are driven by organic matter deposition. For areas such as the Rhone prodelta, continental organic carbon input is dominant, and it is difficult to identify the fraction of labile fraction responsible for fast OM pool consumption (Pastor et al 2011). Moreover, observations show that some organic compounds are preferentially degraded and become selectively oxidized (Middelburg et al., 1997; Pozatto et al., 2018). As a result, the model assumed solid phase organic carbon with three fractions with different reactivities and C/N ratios (Westrich and Berner, 1984; Soetaert et al.,

1996). The mineralization of OM occurs sequentially, with the labile fraction mineralizing faster than the slow decaying carbon. During the timescales considered here, the refractory organic matter class is not reactive. This organic carbon degradation requires oxidants, and the depth-dependency in sequential utilization of terminal electron acceptors assumption first proposed by (Froelich, 1988) is used here. Oxygen is consumed first, followed by nitrate,iron oxides, sulfate and finally methanogenesis occurs (Eq. 3). Because the quantity of organic matter, the relative proportions of fast and slow degrading

materials, and the reactivities decrease with depth, the overall organic matter degradation rate decreases and eventually ceases. In the formulation of the individual biogeochemical processes, we use a similar paradigm as (Soetaert et al., 1996) (Eq. (2)).

This rate of carbon mineralization of organic matter ($mmol\ m^{-3}\ d^{-1}$) can be expressed as:

$$Cprod = \left(rFast \times C_{org}^{fast} + rSlow \times C_{org}^{slow}\right) \times \frac{(1-\phi)}{\phi} \qquad (1)$$

Where the $rFast$ and $rSlow$ are the decay rate constant ($d^{-1}$) for fast and slow detritus component. $\phi$ and $(1-\phi)$ are the

volume fraction for both solutes and solid respectively. This process is mediated by microrganisms and oxidant availabilty. The primary redox reaction includes (1) Oxic respiration (2) Denitrification (3) Fe (III) reduction (4) Sulfate reduction and (5) Methane production:





$$OM + O_2 \rightarrow CO_2 + \frac{1}{(C:N)}NH_3 + H_2O$$

$$OM + 0.8NO_3^- + 0.8H^+ \rightarrow CO_2 + \frac{1}{(C:N)}NH_3 + 0.4N_2 + 1.4H_2O$$

$$OM + 4FeOOH + 8H^+ \rightarrow CO_2 + \frac{1}{(C:N)}NH_3 + 4Fe^{2+} + 7H_2O \qquad (2)$$

$$OM + 0.5SO_4^{2-} + H^+ \rightarrow CO_2 + \frac{1}{(C:N)}NH_3 + 0.5H_2S + H_2O$$

$$OM \rightarrow 0.5CO_2 + \frac{1}{(C:N)}NH_3 + 0.5CH_4$$

This reaction can be modelled using a Monod type relationship with each oxidant having a half-saturation constant ($k_{s[C]}$) represented as $ks^*$ in the model code. The inhibiton of mineralization by the presence of other oxidants is also modelled with a hyperbolic term (subtracted from 1) where $k_{in[C]}$ is concentration at which the rate drops to half of its maximal value. Using these limitation and inhibition functions, a single equation for each component across the model-depth domain can be realized (Rabouille and Gaillard, 1991; Soetaert et al., 1996; Wang and Van Cappellen, 1996), together with some possible overlap (Froelich et al., 1979; Soetaert et al., 1996). For a generic species, this can be described mathematically as:

$$lim = \frac{[C]}{(k_{s[C]} + [C])}\prod\left(1 - \frac{[C]}{(k_{in[C]} + [C])}\right) \qquad (3)$$

Where C is one oxidant. Formulation for individual pathways as well as values of half-saturation and inhibition constants for each oxidant can be found in Appendix (A1). With this limitation term, mineralization rate per solute can be estimated using potential carbon produced via OM degradation in (1):

$$rate_{min} = Cprod \times lim \times \frac{1}{\sum lim} \qquad (4)$$

with the $\sum lim$ the sum of all limitation terms which normalizes the term in order to always achieve the maximum degradation rate. See Soetaert et al. (1996) for more details on the derivative of this equation.

Secondary redox reaction includes reoxidation of reduced substances (nitrification, Fe oxidation, $H_2S$ oxidation, methane oxidation) (Eq. 5) and the precipitation of FeS. Anaerobic oxidation of methane occurs in the absence of $O_2$ following upward diffusion of methane to the sulfate-methane transition zone (SMTZ) (Jørgensen et al., 2019):

$$NH_4^+ + 2O_2 \rightarrow NO_3^- + H_2O + 2H^+$$
$$4Fe^{2+} + O_2 + 6H_{20} \rightarrow 4FeOOH + 8H^+$$
$$CH_4 + O_2 \rightarrow CO_2 + 2H_2O$$
$$H_2S + O_2 \rightarrow SO_4^{2-} + 2H^+ \qquad (5)$$
$$CH_4 + SO_4^{2-} \rightarrow HCO3^- + HS^- + H_2O$$
$$Fe^{2+} + H_2S \rightarrow FeS + 2H^+$$



These reactions are mathematically described using a coupled reaction formulation. Nitrification is limited by the availability of oxygen and the other reaction are described with a first-order term.

$$
\begin{aligned}
Nitri &= R_{nit} \times NH_4 \times \frac{O_2}{(O_2 + ks_{nitri})} &&\text{(Nitrification)}\\
Feoxid &= R_{FeOH_3} \times Fe \times O_2 &&\text{(Iron oxidation)}\\
H2Soxid &= R_{H_2S} \times H_2S \times O_2 &&\text{(sulfide oxidation)}\\
CH4oxid &= R_{CH_4} \times CH_4 \times O_2 &&\text{(Methane oxidation)}\\
AOM &= R_{CH_4} \times CH_4 \times SO_4 &&\text{(Anaerobic oxidation of methane} \quad (6)
\end{aligned}
$$

Where $R_{nit}$, is the maximum rate of Nitrification ($d^{-1}$), $R_{FeOH_3}$, $R_{H_2S}$, $R_{CH_4}$ are the maximum rate of oxidation of Iron, sulfide and methane via oxygen respectively ($mmol^{-1}$ $m^3$ $d^{-1}$). Because sulfide precipitation can occur in some coastal sediments, we accounted for this sink process by removing produced sulfide from sulfate reduction as a first order FeS formation.

$$
\begin{aligned}
FeSprod &= R_{FeSprod} \times Fe \times H_2S &&\text{(FeS production)}\\
H_2S_{oxid} &= rH_2S_{oxid} \times O_2 \times H_2S &&\text{(Sulfide oxidation)} \quad (7)
\end{aligned}
$$

with $R_{FeSprod}$ the rate of production of FeS ($mmol^{-1}$ $m^3$ $d^{-1}$).

### 2.2.3 Transport processes

Transport processes in the model are described by molecular diffusion and bio-irrigation for dissolved species whereas bioturbation is the main process for mixing the solid phase. In addition, Advection occurs in both the solid and dissolved species. The model dynamics described as a partial differential equation (PDE) is the general reaction-transport equation (Berner, 1980). We use a similar paradigm and formulations to that of (Soetaert et al., 1996). For substances that are dissolved:

$$
\frac{\partial \phi C}{\partial t} = -\frac{\partial}{\partial z}\left[-\phi \times D_{sed} \times \frac{\partial C}{\partial z} + w_\infty \times \phi_\infty \times C\right] + \sum \phi \times REAC \quad (8)
$$

With special consideration of ammonium adsorption to sediment:

$$
\frac{\partial \phi C}{\partial t} = -\frac{\partial}{\partial z}\left[-\frac{\phi \times D_{sed}}{(1 + k_{ads})} \times \frac{\partial C}{\partial z} + w_\infty \times \phi_\infty \times C\right] + \sum \frac{\phi \times REAC}{(1 + k_{ads})} \quad (9)
$$

For the solid phase:

$$
\frac{\partial (1 - \phi)S}{\partial t} = -\frac{\partial}{\partial z}\left[-(1 - \phi) \times D_b \times \frac{\partial S}{\partial z} + w_\infty \times (1 - \phi)_\infty \times S\right] + \sum(1 - \phi) \times REAC \quad (10)
$$

where C is the concentration of porewater (unit of mmol $m^{-3}$ liquid) for Eq. (8) and S for solid (unit of mmol $m^{-3}$ solid) Eq. (10). $w$ (cm $d^{-1}$) and $D_{sed}$ (cm$^2$ $d^{-1}$) represent the burial/advection and molecular diffusion coefficient in the sediment





respectively and REAC is the source/sink processes linked to biogeochemical reactions in the sediment. $D_b$ is the bioturbation term for solid driven by the activities of benthic organisms. For dynamic simulation, $w$ can change as a function of time but in most cases we assumed a constant value.

Diffusive fluxes of solutes across the sediment-water interface are driven by the concentration gradients between the overlying seawater and the sediment column. Fick's first law is used to describe the solute flux due to molecular diffusion:

$$J_d = -\phi D_{sed} \frac{\partial C}{\partial z} \qquad (11)$$

where the $D_{sed}$ (cm d$^{-1}$) is the effective diffusion coefficient corrected for tortuosity and given as $D_{sed} = \frac{D^{sw}}{\theta^2}$, with $D^{sw}$ the molecular diffusion coefficient of the solute in free solution of sea-water and $\theta$ is the tortuosity derived from the formation factor ($F$) and porosity ($\phi$) of a sediment matrix (Berner, 1980; Boudreau, 1997). This molecular diffusion coefficient is calculated as function of temperature and salinity using compiled relation of (Boudreau, 1997), implemented in the R package *Marelac* (Soetaert and Petzoldt, 2020).

As a simplifying assumption, material accumulation has no effect on porosity. We further assumed the porosity profile decreased with depth but invariant with time. Although, this assumption is a restrictive as site of flood deposition can undergo variation in grain size which might affect their porosity (Cathalot et al., 2010), we proceed noting that the fixed parameters which define the porosity curve can be changed when necessary. Thus, using optimized parameters fitted with data in the proximal sites of Rhone prodelta (Ait Ballagh et al., 2021), porosity ($\phi(z)$) in Eq. (10) - (8) is prescribed as an exponential decay:

$$\phi(z) = \phi_\infty + (\phi_0 - \phi_\infty)e^{\frac{-(z-z_{swi})}{\delta}} \qquad (12)$$

Where $\phi_0$ and $\phi_\infty$ is the porosity surface and at deeper layer respectively while $Z_{swi}$ is the depth of the SWI and $\delta$ $(cm)$ is the porosity decay coefficient with depth.

### 2.2.4 Bioturbation and Bio-irrigation

Bioturbation in the model is characterized by the movement and mixing of particles by benthic organisms. This is parameterized as a diffusivity function in space ($D(z)$) and acts on the concentrations of the different solid species in the sediment. In our model, this bioturbation flux is assumed to be interphase, with porosity $phi(z)$ remaining constant over time. Thus, this process is prescribed as:

$$Db(z) = \begin{cases} D_b^0 & \text{if } Z \leq Z_L \\ D_\infty + (D_b^0 - D_\infty)e^{-\frac{(Z-Z_L)}{biot_{att}}} & \text{if } Z > Z_L \end{cases} \qquad (13)$$




Where $D_b^0$ is the bio-diffusivity coefficient (cm$^2$d$^{-1}$) at the SWI and in the mixed layer, $Z_L$ is the depth of the mixed layer (cm) and $biot_{att}$ is the attenuation coefficient (cm$^{-1}$) of bioturbation below the mixed layer. $D_\infty$ is the diffusivity at the deeper layer as usually specify as zero. In the model, we did not account for mortality of benthic fauna following the deposition as in De Borger et al. (2021) where they focus on habitat recolonization after trawling.

Bio-irrigation is modelled in an identical manner to that of biodiffusion and acts as a non-local exchange process between the porewater parcels and the overlying bottom water.

$$Irr(z) = \begin{cases} Irr_0 & \text{if } Z \leq Z_L \\ Irr_\infty + (Irr_0 + Irr_\infty)e^{-\frac{(Z-Z_L)}{Irr_{att}}} & \text{if } Z > Z_L \end{cases} \qquad (14)$$

for which $Irr_0$ is the bio-irrigation rate (d$^{-1}$) and $Irr_{att}$ is the attenuation of irrigation ($cm$) below the depth of the irrigated layer $Z_{irr}$ (cm). At depth, the bio-irrigation rate ($Irr_\infty$) is generally set to zero.

### 2.2.5 Model vertical grid

The model is vertically resolved with grid divided into 100 layers ($N$), of thickness ($\Delta z$) increasing geometrically from 0.01 cm at the sediment-water interface to 6 cm at the lower boundary. The result is a 100 cm model domain comprising of a full grid with non-uniform spacing and maximum resolution near the SWI. Depth units are in centimeters. This choice of modelled depth allows for complete carbon degradation. During the time instance of the event specification, the added grid of new layers ($N_{pert}$) and the current grid ($N_{cur}$) is rescaled to the model's common grid of $N$ layer by linear interpolation (see section 2.2.6 and Fig. S1). The concentration of state variables is defined at the layer midpoints, whereas diffusivities, advection (sinking/burial velocities), and resulting transport fluxes are defined at the layer interfaces.

### 2.2.6 Deposition event

The inclusion of the deposition event as a separate external routine to modify the sediment properties (i.e porewater species, $C_{org}$) is a fundamental difference between our approach and the other previous early diagenesis model applied in the Rhone Delta, but it is similar to De Borger et al. (2021). We assume the event occur as an instantaneous deposition of organic carbon ($C_{org}^{fast}$ and $C_{org}^{slow}$) over a depositional layer ($Z_{pert}$). Furthermore, we assume that the solid and solute components are homogenously mixed over the deposited layer, with the solutes ($O_2$, $NO_3^-$, $NH_4^+$, DIC, $SO_4^{2-}$) set to the bottom water concentration (Fig. 2).

The event calculation was carried dynamically within the same time run. Following the flux of organic carbon via the boundary condition (see section 2.2.7), the portion of organic carbon is split between the fast and slow decaying component using a proportionality constant ($pfast$) as in Ait Ballagh et al. (2021). However, at the time when the event is prescribed, the integrated profile of the solid species $C_{org}^{fast}$ and $C_{org}^{slow}$ from previous time step ($t^-$), was used to create a virtual composite of





the deposited layer. This integral calculation was performed over a specified sediment thickness ($Z_{pert}$), which corresponded

to the extent of the depositional event. This average concentration for the solid ($C_{org}^{flood}$) scaled with an enrichment factor ($\alpha$

see below) was then nudged on top of the old layer which is supposed to be buried beneath after the event. For the solutes, the

bottom water concentration is imposed through the perturbed layer at the time of event. To avoid numerical issues caused by

the discontinuity of both layers with different properties, a staggered interpolation of the composite profile was performed over

the modeling domain. This smoothes the interface between the deposited layer's base and the current model grid's upper layer.

This algorithmic procedure is schematically shown in Figure. 2 and we summarized this process mathematically as:

$$C_{org}^{flood} \approx \alpha C_{org}^i(t)\big|_0^{z_{pert}} = \alpha \frac{\int_0^{Z_{pert}} C_{org}^i(t^-)dz}{Z_{pert}} \begin{cases} \alpha < 1, \text{ if } TOC_{z_{pert}} < TOC_{old} \\ \alpha > 1, \text{ if } TOC_{z_{pert}} > TOC_{old} \end{cases}; \quad i = 1(fast), 2(slow) \quad (15)$$

The carbon enrichment factor ($\alpha$) denoted as *confac* in the model code is introduced in order to scale the deposited OC with

those observed from field data. This helps in calibrating the deposited organic matter concentration ($C_{org}^{fast}$ and $C_{org}^{slow}$) in the

new layer relative to the previous sediment fraction, simulating the wide range of TOC content observed in the field. For

instance, when the newly deposited organic matter is similar to the former sediment topmost layer (average preflood layer

concentration over an equivalent $Z_{pert}$ depth), an $\alpha$ value of 1:1 is used. If the new material is lower in organic carbon content

compared to what is near the sediment-water interface, then $\alpha < 1$, while if the newly deposited material is higher in carbon

content than the sediment surface, $\alpha > 1$. This flexibility can be used to constrain the simulation to match the corresponding

TOC profile from field observation.

In modeling application, this parameter is generally specified by using different value for the magnitude of OC in each fraction

depending on the empirical observation of the TOC data. This quantity is therefore tunable and the upper bound of this

parameter is dictated by the maximum TOC in the sediment sample.



*Geoscientific*
*Model Development*
*Discussions*

EGU

*Figure 2: Schematic of model implementation for the deposition event scenario. Profile from previous time step (left) and after addition of new layer over a predefined depth layer (right). For the solid (A), the new layer can be enriched (blue) or depleted (red) relative to the old (average) (black). The dissolved substance (B) are set equal to the bottom water concentration during the deposition. Thereafter, the profile is integrated forward with time. The whole sequence of step occurs dynamically with time capitalizing on the integrator ability to simulate dynamic event process. α is the carbon enrichment factor applied over depth $Z_{pert}$ (see text for detail).*

### 2.2.7 Boundary Conditions

The boundary conditions for the model are of three type:





- At the sediment-water interface, a Dirichlet concentration condition for most solutes equaling the bottom water concentration was used.


$$C|_{z=0} = C_{bw} \qquad (17)$$

- Both pore water and solid have a zero-flux boundary condition at the bottom of the model:

$$\frac{dC}{dz}|_{z=z_\infty} = 0 \qquad (18)$$

- For solid, an imposed flux at the upper boundary for most of the year is used:

$$flux_{org}|_{z=0} = -\left(1 - \phi_0\right)Db_0\frac{dC}{dz}|_{z=0} + \left(1 - \phi_\infty\right)w_\infty C|_{z=0} \qquad (19)$$

The model also includes the ability to include time-varying organic carbon flux with user-specific time-series or a functional representation such as sinusoidal pattern. For the latter case, we used a yearly carbon flux ($\overline{flux}_{org}$) with maximum value in the spring and lower flux in fall and winter. This was expressed mathematically as:

$$flux_{org}(t)|_{z=0} = [\overline{flux}_{org} \times (1 + sin(\frac{2 \cdot \pi \cdot t}{365}))] \qquad (20)$$

At the time of the instantaneous deposition, this deposited carbon is treated as described in section 2.2.6.

**2.2.8    Model parameterization and verification**

The model parameters in Table. 2 (for full model parameter see Table S1 in supplementary) were derived from previously published model in the Rhone Delta (Pastor et al., 2011; Ait Ballagh et al., 2021). The organic matter stoichiometry for both fractions is represented here by the NC ratio (*NCrFdet* and *NCrSdet*) with value of 0.14 and 0.1 respectively. The flux of carbon in the upper boundary of the model was defined using a yearly mean flux ($\overline{flux}_{org}$) of 150 mmol m$^{-2}$ d$^{-1}$ in Rhone

prodelta (Pastor et al., 2011). Model derived total organic carbon (TOC) is estimated from both carbon fraction ($C_{org}^{fast}$ and $C_{org}^{slow}$) as:

$$TOC = \left(C_{org}^{fast} + C_{org}^{slow}\right) \times 1200 \times \frac{10^{-9}}{2.5} + TOC_{ref} \qquad (21)$$

with $TOC_{ref}$ the asymptotic TOC value at deeper layer of the sediment, thus representing concentration of refractory carbon not explicitly modelled. The sedimentation rate used in this modelling study was kept constant at 0.03 cm d$^{-1}$ (Pastor et al.,

2011). The decay rate constant for the labile and semi-labile detritus matter is set as 0.1 and 0.0031 d$^{-1}$ respectively with both





fractions split equally with a proportionality constant ($pfast$) of 0.5. Using parameters fitted by the model of Ait Ballagh et al. (2021) to data observed in the Rhone prodelta area, the rate of bioturbation and bio-irrigation is fixed as 0.01 cm$^2$d$^{-1}$ and 0.23 d$^{-1}$ with these fauna-induced activities occurring down to a depth of 5 and 7 cm respectively.

The bottom water temperature was fixed at 20°C. The bottom water salinity is nearly constant below the Rhone River plume,
ranging from 37.8 to 38.2. In the model, the average temperature and salinity is used to calculate the diffusion coefficient for the solute chemical species (section 2.2.3). Bottom water solute concentrations were constrained using previously reported values in previous modelling efforts (Ait Ballagh et al., 2021) and adapted with new data for the time corresponding to the flood deposit event (see Table 2). Porosity decreases exponentially with depth from 0.9 at the sediment water interface to 0.5 at deeper layer with a decay coefficient of 0.3 cm (Lansard et al., 2009).

Table 2: Core parameters used in the model

| Model Parameters | Model Notation | Values | units | description | References |
|---|---|---|---|---|---|
| $\overline{flux}_{org}$ | CFlux | 150 | $mmol\ m^{-2}d^{-1}$ | total organic C deposition | Pastor et al., 2011 |
| $pfast$ | pFast | 0.5 | - | part FDET in carbon flux | Pastor et al 2011 |
| $\overline{flux}_{FeOO3}$ | FeOH3flux | 0.01 | $mmol\ m^{-2}d^{-1}$ | deposition rate of FeOH3 | Assumed |
| rFast | rFast | 0.1 | $d^{-1}$ | decay rate FDET | Ait Ballagh et al., 2021 |
| rSlow | rSlow | 0.0 | $d^{-1}$ | decay rate SDET | Ait Ballagh et al., 2021 |
| NCrFdet | NCrFdet | 0.1 | $molN/molC$ | NC ratio FDET | Pastor et al 2011 |
| NCrSdet | NCrSdet | 0.1 | $molN/molC$ | NC ratio SDET | Pastor et al 2011 |
| $O_{2_{bw}}$ | O2bw | 197 | $mmol\ m^{-3}$ | upper boundary O2 | Ait Ballagh et al., 2021 |
| $NO_{3_{bw}}$ | NO3bw | 0.0 | $mmol\ m^{-3}$ | upper boundary NO3 | Ait Ballagh et al., 2021 |
| $NH_{3_{bw}}$ | NH3bw | 0.0 | $mmol\ m^{-3}$ | upper boundary NH3 | Ait Ballagh et al., 2021 |
| $CH_{4_{bw}}$ | CH4bw | 0.0 | $mmol\ m^{-3}$ | upper boundary CH4 | Rasmann et al., 2016 |
| $DIC$ | DICbw | 2360 | $mmol\ m^{-3}$ | upper boundary DIC | Pastor et al., 2018 |
| $Fe^{2+}_{bw}$ | Febw | 0.0 | $mmol\ m^{-3}$ | upper boundary Fe2 | Pastor et al., 2018 |
| $H_2S_{bw}$ | H2Sbw | 0.0 | $mmol\ m^{-3}$ | upper boundary H2S | Pastor et al., 2018 |
| $SO_{4_{bw}}$ | SO4bw | 30246 | $mmol\ m^{-3}$ | upper boundary SO4 | Pastor et al., 2018 |
| $w$ | w | 0.027 | $cm\ d^{-1}$ | advection rate | Pastor et al 2011 |
| $D_0$ | biot | 0.01 | $cm^2\ d^{-1}$ | bioturbation coefficient | Ait Ballagh et al., 2021 |





| Model Parameters | Model Notation | Values | units | description | References |
|---|---|---|---|---|---|
| $Z_L$ | biotdepth | 5 | $cm$ | depth of mixed layer | Ait Ballagh et al., 2021 |
| $biot_{att}$ | biotatt | 1.0 | $cm^{-1}$ | attenuation coeff below biotdepth | Ait Ballagh et al., 2021 |
| $Irr_0$ | irr | 0.2 | $d^{-1}$ | bio-irrigation rate | Ait Ballagh et al., 2021 |
| $Z_{irr}$ | irrdepth | 7 | $cm$ | depth of irrigated layer | Ait Ballagh et al., 2021 |
| $Irr_{att}$ | irratt | 1.0 | $cm$ | attenuation coeff below irrdepth | Ait Ballagh et al., 2021 |
| $temp$ | temperature | 16 | ℃ | temperature | Ait Ballagh et al., 2021 |
| $sal$ | salinity | 38 | $psu$ | salinity | Ait Ballagh et al., 2021 |
| $TOC_{ref}$ | TOC0 | 1.1 | % | refractory Carbon conc | Pastor et al., 2018 |
| $\emptyset_0$ | por0 | 0.8 | - | surface porosity | Ait Ballagh et al., 2021 |
| $\emptyset_\infty$ | pordeep | 0.6 | - | deep porosity | Ait Ballagh et al., 2021 |
| $\delta$ | porcoeff | 2 | $cm$ | porosity decay coefficient | Ait Ballagh et al., 2021 |

For the verification of the model output, data from (Pastor et al., 2018) corresponding to the diagnetic situation 26 days after an organic-rich flood were used. We restricted our benchmark to data from the proximal station (Station A) near the river mouth, where the impact of this flood discharge is more visible (Fig. 1).

### 2.2.9    Numerical Integration, Application & Implementation

Equation (8)-(10) are integrated numerically over the model grid and time duration. The model timestep ($dt$) is set by the user and is generally problem specific. Because of the aforementioned challenge in observability of the massive flood event deposition, daily resolution is most often used for $dt$. However, there is a possibility of obtaining higher resolution by decreasing $dt$.

The model application starts by estimating the steady-state condition of the model using the high level command *FESDIAperturb()*. This steady-state condition is then used as a starting condition for a dynamic simulation, with perturbation times as in *perturbTimes* and depth of perturbation given as *perturbDepth* in the model function call. As the event can be given as a deposit and mixing process, further specification of the perturbation type (*deposit* and *mix*) is provided as an argument to the simulation routine. In our case, we used only the deposit mode. The event algorithm is used at the stated time point to estimate the model pore-water and solid properties driven by the instantaneous change in the boundary





condition. The concentrations are successively updated by their diagenetic contributions during this time step. Afterward, this modified profile is integrated forward in time. The model is written in Fortran for speed and integrated using the R programming language (R Core Team, 2021) via the "method-on-lines" approach (Boudreau, 1996). In addition, the model made use of the event-handling capabilities specific numerical solvers written in the R deSolve package (Soetaert et al., 2010b).

Preprocessing routine for model grid generation (Soetaert and Meysman, 2012), porewater chemistry parameter (Soetaert et al., 2010a), steady state calculation (Soetaert, 2014), and time integration (Soetaert et al., 2010b) were utilizes the R programming language. Further information about the model usage can be found in the model user vignette found in R-forge page (https://r-forge.r-project.org/R/?group_id=2422).

### 2.2.10 Quantification of sediment diagenetic relaxation timescale

•    Quasi-relaxation timescale

Given the strong non-linearity and coupled nature of the biogeochemical system in question, we used an approximate approach to define the timescale of relaxation. Recognizing that in a nonlinear system, a perturbed trajectory is frequently arbitrarily divided into a fast, transient phase and a slow, asymptotic stage that closes in on the attractor (i.e. steady state concentration; Kittel et al., 2017), we proceeded to estimate the relaxation timescale by using the time for which the memory of the perturbed

signature disappears. We estimate the relaxation timescale by first calculating the absolute difference ($\varphi(t)$) between successive model output after the event, assuming that a slow stationary state will eventually converge to the pre-perturbed state as time after the disturbance approaches infinity. This point-by-point concentration difference between two successive discretized profiles is then terminated at the point where the sum of absolute differences at each time point is less than the threshold (i.e given by the median over the entire time duration). The relaxation time, $\tau$ for each porewater profile species is

then defined as the first time this threshold is crossed. Similar technique was employed by (Rabouille and Gaillard, 1990).

$$\varphi(t) \quad = \frac{1}{N} \sum_{i=1}^{N} \| X_{t+1}^i - X_t^i \|$$

In the limit of time (t):

$$\tau(t) \quad \Rightarrow \overline{\varphi(t)} \approx \text{seasonal background} \qquad (22)$$

Where N is the total number of grid point ($i$) used to discretized the depth profile ($X_t$) and $X_{t+1}$ is the depth profile at $t+1$ after the event.

•    Uncertainty in relaxation timescale estimate

The uncertainty introduced by this technique is quantified using a non-parametric bootstrap of the $\tau$ statistics. The objective of bootstrapping is to estimate a parameter based on the data, such as a mean, median, or any scalar or vector statistics but with less restrictive assumptions about the form of the distribution that the observed data came from (Efron, 1992).





In this case, we employ a modified bootstrapping technique to estimate the cutoff point based on the median of the $\overline{\varphi}$ in a given reference simulation. The variance of this reference's background over time indicates the uncertainty of this median cutoff point. This reference median and its variance are then simulated *n times* to generate random perturbations around the normative median value. We can create a histogram of the replicate distribution based on these runs, and the relaxation time in each iteration is calculated ($\hat{\tau}$). The median absolute deviation from this ensemble of relaxation times is then used to calculate the level of uncertainty in the statistics of interest (timescale of relaxation - $\hat{\sigma}_{\tau}$). Figure 3 depicts this concept schematically.




*Figure 3: The bootstrapping technique used to calculate the uncertainty in the relaxation timescale. The resampled median about a reference provides a replicate over which the standard error estimate is defined. The solid red represents the expected value of the quantity estimated while the vertical red line is the deviation from this expected value.*

The 95 % confidence intervals ($\hat{\sigma}_{\tau(level/2)}$ and $\hat{\sigma}_{\tau(1-level/2)}$) are reported in this paper by calculating the quantiles of this

empirical distribution of $\hat{\sigma}_{\tau}$.





### *2.2.11* **Model simulation**

The model is initialized as explained in 2.2.9. Thereafter, for the dynamic simulation, the model is spin-up for a sufficiently long-time to attain dynamic equilibrium (≥ 5 years). A two-year run is carried out for the respective model application. The timestep ($dt$) for dynamic simulation is daily in order to match the frequency for which observation of field data is possible.
For specific numerical experiment, model configuration required for the simulation will be detailed in section 2.2.11.1.

#### 2.2.11.1 *End-member type numerical experiment*

For the numerical model experiment, we investigate the sediment's response to two end member types of deposition that can represent actual field observations in the Rhone prodelta (Pastor et al., 2018).

- **Low OC content with high sediment thickness scenario (EM1):** In this scenario, we assume that a 30 cm new layer
of sediment of degraded sediment was deposited. This scenario can describe old terrestrial material and is similar to the extreme case of flood event of May/June 2008 in the proximal outlet of Rhone River where lateral transfer of low TOC sediment (around 1%) was deposited on top of the previously deposited sediment (OC around 1.5-3%) (Cathalot et al., 2010). Using the partitioning of the carbon as explained in section 2.2.7, An $\alpha$ value of 0.5 and 0.7 for $C_{org}^{fast}$ and $C_{org}^{slow}$ respectively was used to scale the TOC profile in order to mimic this type of trend.

- **High OC with low sediment thickness scenario (EM2):** For this, we assumed a moderate 10 cm deposition of a new layer enriched in carbon during a flood discharge event. This scenario can correspond to the end-member case of November 2008 flood type with high TOC around 2.5%, reaching more than 6% in some sediment cores from the prodelta (Pastor et al., 2018), (most likely composed of freshwater phytoplankton detritus, debris and freshly dead organisms) overlain on a less labile layer. In order to simulate this type of pattern, an $\alpha$ value of 20 and 10 for $C_{org}^{fast}$
and $C_{org}^{slow}$ respectively was used to adjust the TOC profile to such high deposit OC scenario.

The time of the event occurrence in both scenarios were initialized at period corresponding to published date for May and November 2008 flood deposition as reported in (Pastor et al., 2018). This helps to provide some realism to this hypothetical case study as well as appropriate context to the environmental regime when these events occurs.

#### 2.2.11.2 *Sensitivity analysis*

Lastly, we conducted a sensitivity analysis of the relaxation timescale for oxygen, sulfate, and DIC concentrations in terms of their variation to the thickness of the new sediment layer as well as the quantity of organic carbon introduced by the deposition.

We assumed a 15 cm average deposit thickness and conducted simulations with a thickness scale of 1 cm to 30 cm in 5 cm increments. The $\alpha$ value is calculated in the same way: assuming a 1:1 ratio in the fast and slow OC fractions, and because





deposited sediment can be highly refractory in nature, we geometrically conducted simulations with values ranging from 0.3
to 35. We also made sure that both series are equilateral in length, and that the values were chosen to span the range of values
in EM1 and EM2, thus bracketing the normative value for the end-member case. This range encompasses the large spectrum
of flood deposits such as those experienced in the Saguenay Fjord, Canada (Deflandre et al., 2002; Mucci and Edenborn, 1992),
the Rhone prodelta, France (Pastor et al., 2018), and in the Po River, Italy (Tesi et al., 2012).

## *3*    **Results**

### *3.1*    **Qualitative model performance: Cevenol flood in the Rhone prodelta**

In order to compare the model evolution to field data, we made a comparison between the simulated profiles 26 days after a
flood layer deposition and data collected in the Rhone prodelta in December 2008 (observed data collected 26 days after a
cevenol flood). During this flooding period, riverine discharge delivered $0.4 \times 10^6$ t of sediment which amounted to
approximately 10 cm of sediment deposited in the site A of depo-center (Fig. 4).

The general pattern of the simulated profile agrees well with the observed data (Fig. 4). The newly introduced organic carbon-
rich sediment resulted in rapid oxygen consumption. The data for Total organic carbon (TOC) shown in Figure 4 suggests a
good agreement with the model, with high TOC (2.5 - 2.0 wt%) deposited at the upper 10 cm. 26 days after the flood, the
oxygen concentration dropped from 250 µM at the new sediment interface to nearly zero at 0.2 cm depth, and oxygen may
have already returned to pre-flood levels; the simulated porewater profile was within the data's range (Fig. 4). The model
diffusive flux of oxygen at this period was 18 mmol m$^{-2}$ d$^{-1}$ while the measured DOU flux was $16.6 \pm 2.9$ mmol m$^{-2}$ d$^{-1}$.

Overall the Model-Data trend was satisfactory with observed depth distribution of sulfate ($SO_4^{2-}$) 26 days after the flood event
fitted well, without much parameter fine-tuning. Sulfate reduction was high in the new layer. However, below the flood layer,
the $SO_4^{2-}$ concentration in the data seem to asymptote to a value of 10 mM at 25 cm, while the model simulates complete
sulfate depletion below 20 cm (Fig. 4).





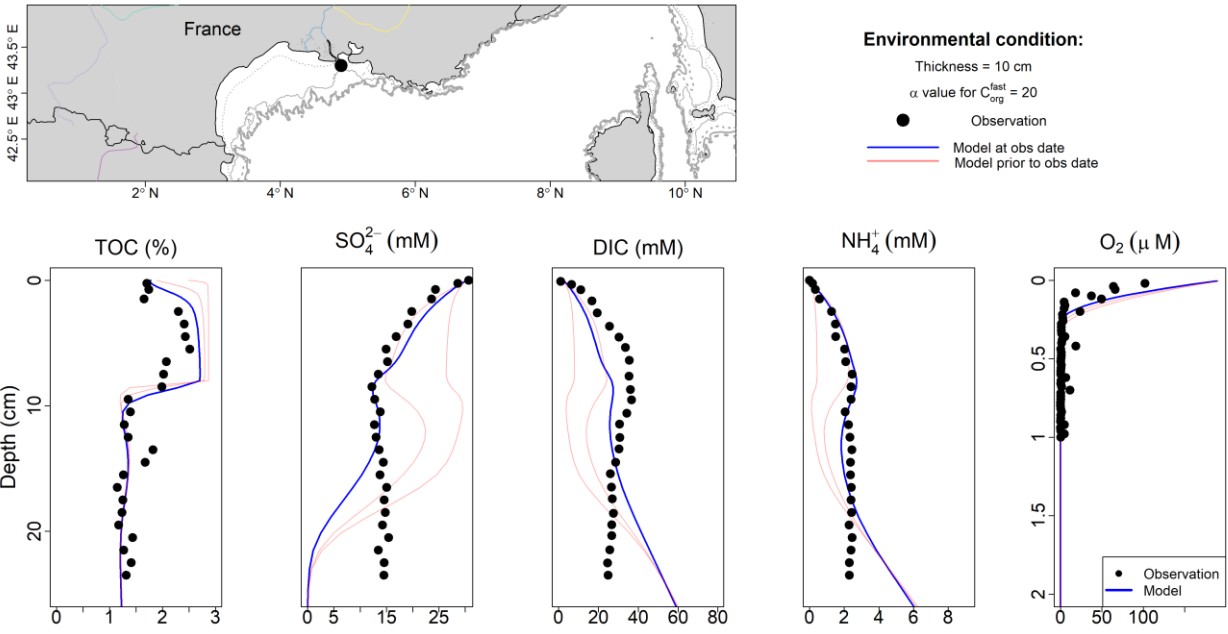


*Figure 4: Model and observation depth profile of TOC (%), $SO_4^{2-}$, DIC, $NH_4^+$ and $O_2$ for November/December Event in Station A (Rhone Prodelta). Model results are at 26 days after flood event in blue line. Data collected in December 8, 2008 showed in black circle. The red lines represent the evolution of the model from the deposition (day - 0 and 10).*

The dissolved DIC profile shows a similar trend to the data collected after the flash flood. Within the depth interval of data,
the model tends to follow the data. It drifts at lower depths, on the other hand, by overestimating the concentration of DIC observed at deeper layers. Similarly, the modeled $NH_4^+$ shows a gradual increase with depth, and the model overestimates the production of $NH_4^+$ below 15 cm (Fig. 4).

### 3.2 Numerical experiment on end-member scenarios

### 3.2.1 Low carbon, High Thickness scenario (EM1)

With a test case of 30 cm of new material deposited during the event (EM1) in the spring, the sediment changes as thus: Prior to the event, the oxygen penetration depth (OPD) was about 0.17 cm. The OPD increases to 1.17 cm after the deposition of these low OC materials. The model showed a gradual return to its previous profile within days, with the OPD shoaling linearly with time (Table. 3). By day 5, oxygen has return to the pre-flood profile with similar gradient to the pre-flood state.




Table 3: Model vs data comparison for oxygen penetration depth (OPD), flux of oxygen, sulfate and DIC (26 days after deposition).

| Time (days) | OPD (cm) | $O_2$ flux (mmol $O_2$ m$^{-2}$ d$^{-1}$) | $SO_4^{2-}$ flux (mmol $SO_4^{2-}$ m$^{-2}$ d$^{-1}$) | DIC flux (mmol DIC m$^{-2}$ d$^{-1}$) |
|---|---|---|---|---|
| **Observation - Model** | | | | |
| Measured | $0.16 \pm 0.03$ | $16.6 \pm 2.9$ | - | - |
| Simulated | 0.2 | 18 | 142 | -203 |


Against a background OM flux following the introduction of the flood layer, the sediment responded quickly. As a result, the perturbation has a significant effect on sulfate penetration depth, with concentration remaining nearly constant within the perturbed depth ($\approx 20$ cm). This corresponds to the bottom water concentration (30 mM) trapped within the flood deposit. Within that layer, sulfate reduction rate was low with an estimated integrated rate of 2.14 mmol C m$^{-2}$ d$^{-1}$ from the surface
to 30 cm.

Below the interface with the newly deposited layer, the sediment is enriched in OM whose mineralization results in a higher sulfate reduction rate (SRR) at the boundary that delineates the newly deposited layer and the former sediment-water interface. The simulated SRR falls from 437 mmol C m$^{-3 \text{ solid}}$ d$^{-1}$ at the former sediment-water interface (now re-located at 26 cm) to 24 mmol C m$^{-3 \text{ solid}}$ d$^{-1}$. This high interior sulfate consumption at the boundary correlates well with the higher proportion of
reactive organic material buried by the new layer containing less reactive material. From day 10, the consumption of this OM stock by sulfate controls the shape of the profile (Fig. 5). This anoxic mineralization via sulfate reduction will continue until the entire stock of carbon is depleted 50 days after deposition. Following that, OM mineralization via sulfate reduction shift becomes more intense at the top layer by day 60 (two months after the event), when it begins to gradually evolve to the typical depth decreasing sulfate profile. By day 115 (~ 4 months), the profile had almost completely returned to its pre-flood state.
We estimate that it took approximately 4 months for sulfate to relax back to within the range of background variability (with lower and upper bootstrap estimate between 92 - 139 days).

Correspondingly, OC mineralization products (such as DIC) were significantly lower in the upper newly deposited layer, as a consequence of the reduced quantity of OC brought by the flood. This concentration increased with depth to about 80 $mM$. Starting from the deposition, higher production of DIC below the former SWI led to a distinct boundary in the sediment: a
DIC depleted layer above an increasing DIC with concentrations up to 75 mM trapped in the region below the new-old sediment horizon 20 days after deposition (Fig. 6). This increased DIC production continued despite complete exhaustion of buried labile fraction with mineralization driven by the slow decaying component. The depth gradient caused by the increased DIC



production enhances diffusive DIC flux. Following that initial period, DIC began to revert to its previous state. This slow re-organization, mostly driven by diffusion continues, with an estimated recovery time of 5 months (with a 95 % bootstrap

confidence interval of 137 - 147 days respectively), as it temporarily lags behind $SO_4^{2-}$ in its return to the previous pre-flood state.

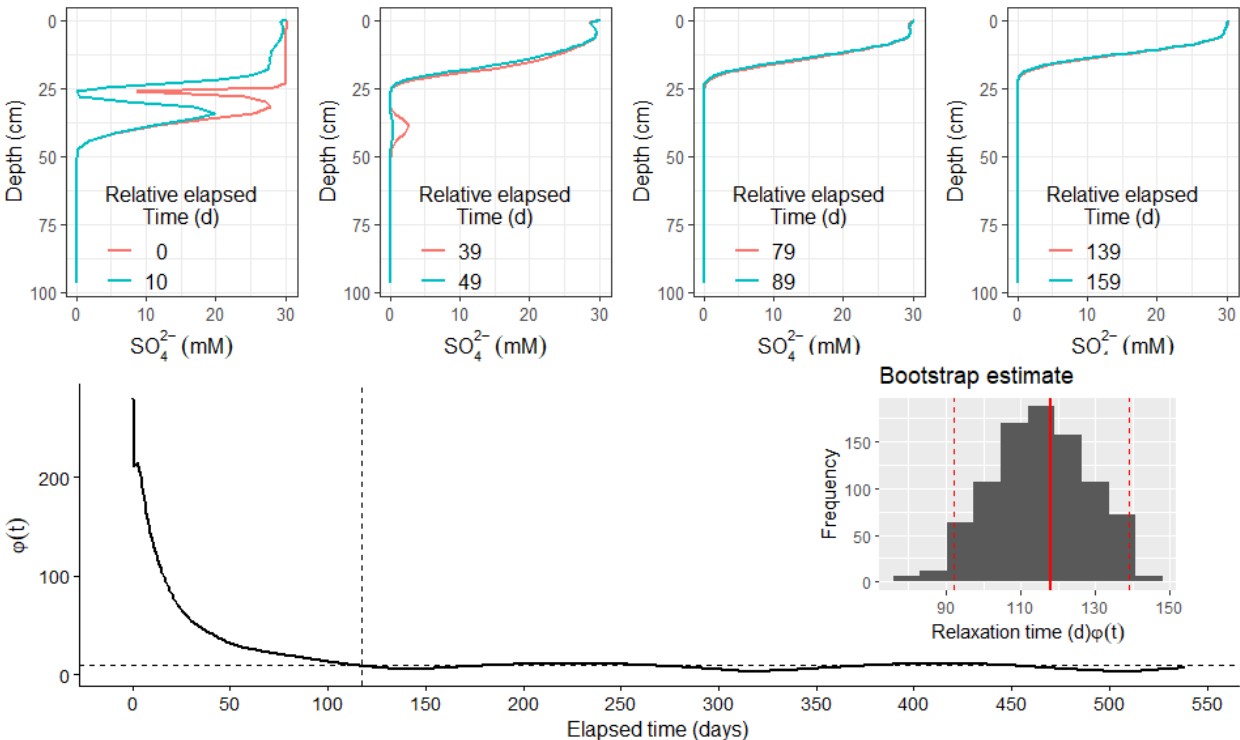

*Figure 5: Scenario 1 (EM1): Model evolution for sulfate following deposition. Relative deviation of successive profile with time shown below. Dashed vertical line signify cutoff point by the median (Dashed horizontal line). Inset: Histogram of bootstrap estimate of sulfate*

*relaxation timescale for EM1 with 95 % confidence interval.*



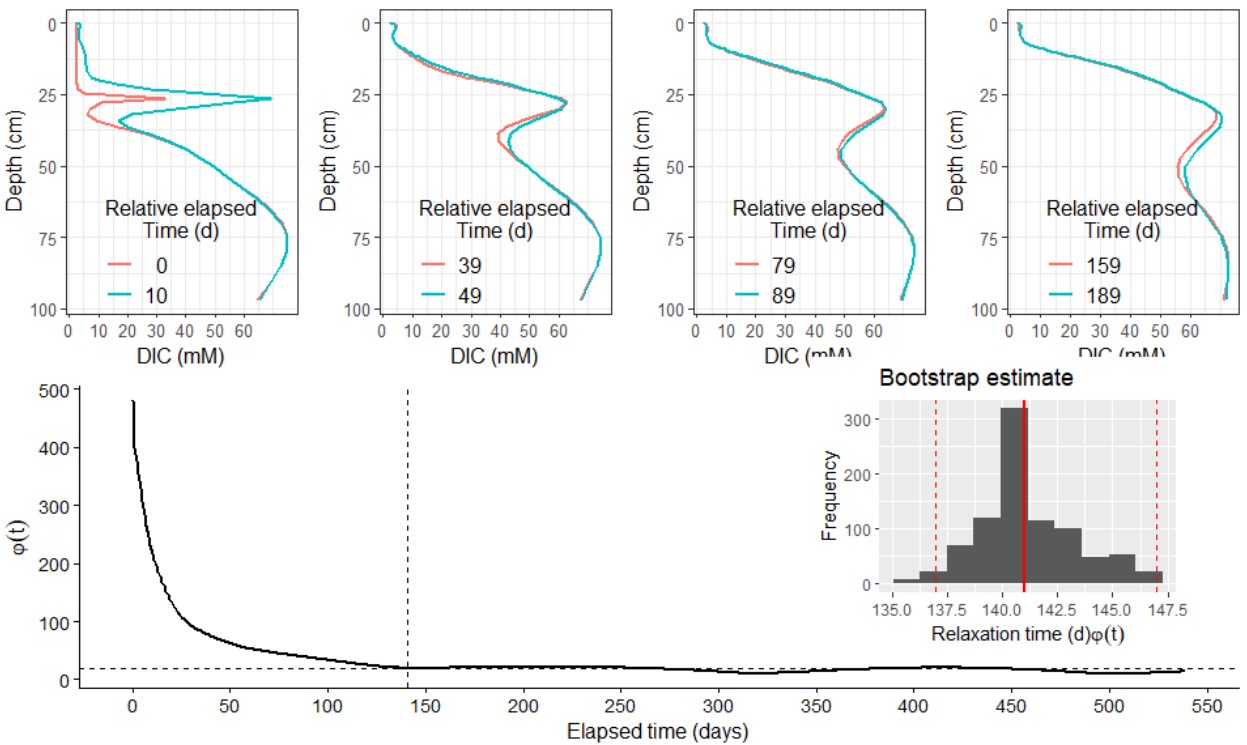

*Figure 6: Scenario 1 (EM1): Model evolution for DIC following deposition. Relative deviation of successive profile with time shown below. Dashed vertical line signifies cutoff point by the median (Dashed horizontal line). Inset: Histogram of bootstrap estimate of DIC relaxation timescale for EM1 with 95 % confidence interval.*

### 3.2.2 High carbon, Low Thickness scenario (EM2)

A flood deposition scenario of 10 cm thick material with enhanced OC content was used for the other end-member case experiment (EM2) in autumn. In this scenario, the modelled sediment exhibits a variety of response characteristics. The newly introduced sediment resulted in rapid oxygen consumption. The OPD decreased to 0.74 cm shortly after the event, according to the model, and stabilized there for days. There was no visible deformation in the shape of oxygen during its recovery trajectory, and total oxygen consumption for organic matter mineralization decreased by 8 % during the first two days after the event, from 12 to 11 mmol $O_2$ m$^{-2}$ d$^{-1}$.

The $SO_4^{2-}$ concentration that developed as a result of the deposition showed two gradients: A concentration gradient from 30 mM at the "new" sediment water interface to 26 mM in the newly deposited layer (Fig. 7). Accordingly, the DIC in the corresponding depth layer gradually increased up to 20 mM (Fig. 8). An intermittent increase in $SO_4^{2-}$ was simulated below the new interface, at the boundary with the "old" sediment-water interface (SWI), reaching up to 29 mM from 9 cm to 12 cm (Fig. 7). This layer, which corresponded to the depth horizon where the new layer gradually mixed with the old layer, resulted





in less sulfate reduction and DIC production in comparison to the new layer. Porewater $SO_4^{2-}$ concentrations decreased monotonically with depth from this interface, with a corresponding increase in DIC. Within 26 days of the event, the sulfate profile appears to be returning to its original shape. By then, 75 % of the newly introduced fraction of OM had been depleted, with OM remineralization in the upper layer fueled by the small amount of remaining detrital materials. As the temporal memory of the deposition fades, the profile continues to gradually evolve towards the background, fed by the slow decaying OM, up to day 90, when the sulfate profile appears to have reached a similar pre-flood state. In this scenario, the estimated $SO_4^{2-}$ and DIC relaxation timescales were around 3 months (91 days for $SO_4^{2-}$ and 102 days for DIC) (Fig. 7).

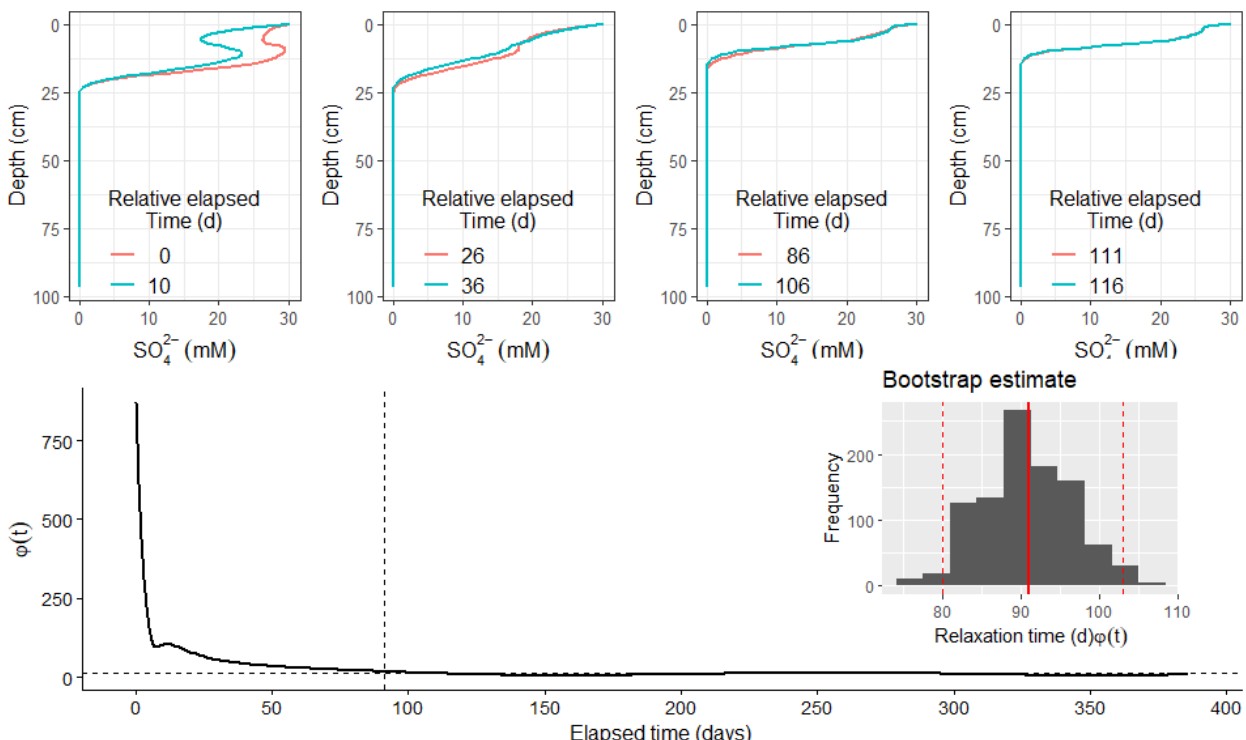

Figure 7: Scenario 2 (EM2): Model evolution for sulfate following deposition. Relative deviation of successive profile with time shown below. Dashed vertical line signifies cutoff point by the median (Dashed horizontal line). Inset: Histogram of bootstrap estimate of sulfate relaxation timescale for EM2 with 95 % confidence interval.

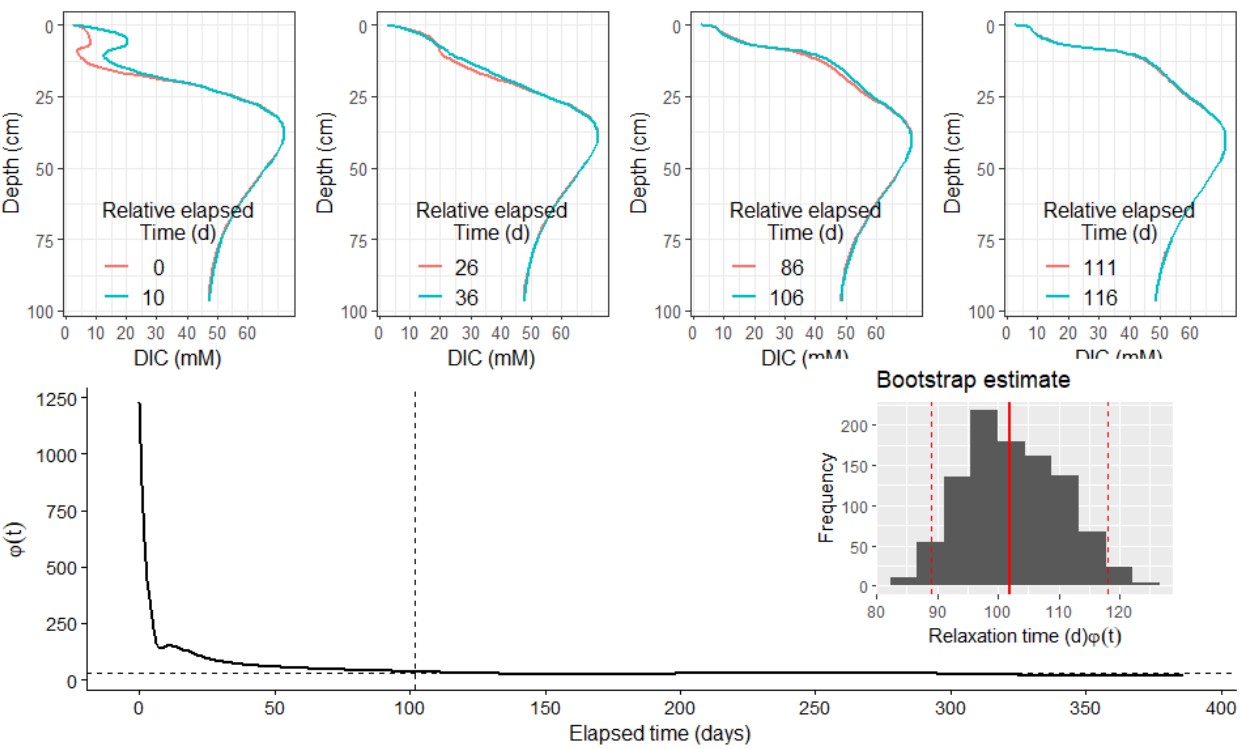

*Figure 8: Scenario 2 (EM2): Model evolution for DIC following deposition. Relative deviation of successive profile with time shown below.*

*Dashed vertical line signify cutoff point by the median (Dashed horizontal line). Inset: Histogram of bootstrap estimate of DIC relaxation timescale for EM2 with 95 % confidence interval.*





### 3.3 Sensitivity of relaxation time to variation in enrichment factor ($\alpha$) and sediment thickness ($z_{pert}$)

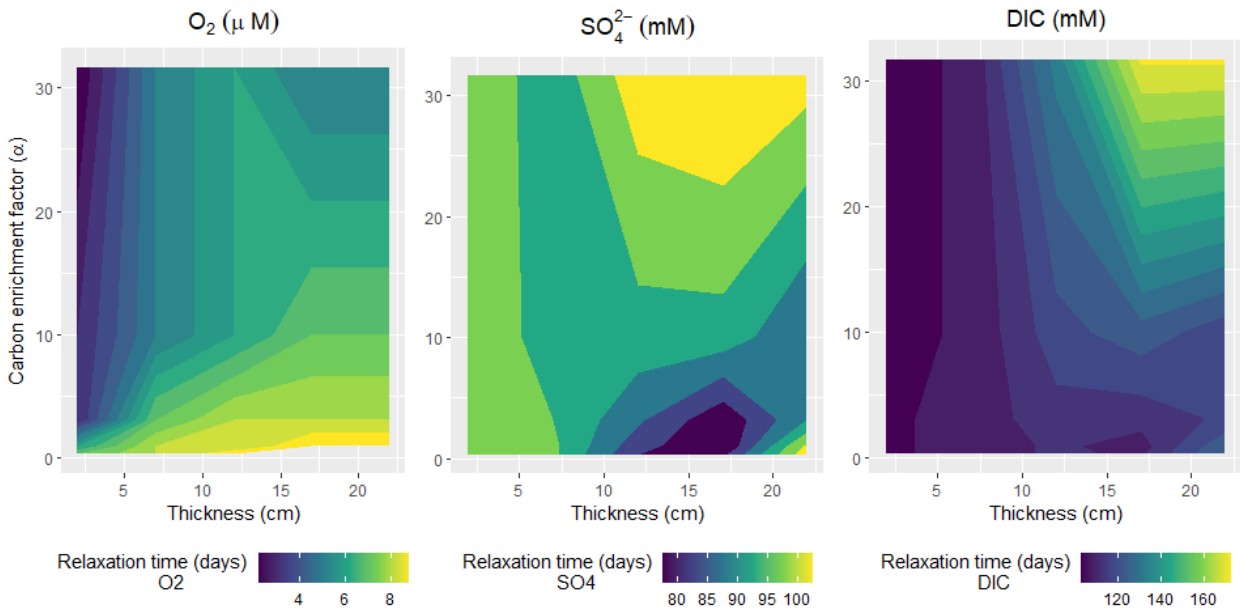

*Figure 9: Relaxation timescale ($\tau$) in days as function of deposited sediment thickness and enrichment factor ($\alpha$) for degradable OM.*

We then examine the sensitivity analysis of the relaxation timescale ($\tau$) for oxygen, sulfate, and DIC for variation in sediment deposit thickness ($Z_{pert}$) and the concentration factor for $C_{org}^{fast}$ enrichment ($\alpha$) covering values ranging between the two EM scenarios.

Over all runs varying the enrichment factor ($\alpha$) and the thickness of the flood input layer, relaxation time for oxygen varied from 2 days for a flood deposited layer consisting of a thin layer of high concentration of labile OC to 9 days for a thicker

deposited layer with low concentration of labile OC. In contrast, the relaxation timescales for $SO_4^{2-}$ and DIC were significantly longer than those for $O_2$ (3 to 4 months). In addition, the relaxation timescale surface structure for $SO_4^{2-}$ appears complex with divergence gradient at mid-depth of 15 cm. For deposited depth layers above 5 cm and at low $\alpha$ value, the relaxation time for $SO_4^{2-}$ varied between 75 - 100 days (2 - 3 months). Below 5 cm (bioturbated depth imposed in the model), relaxation time was constant across all $\alpha$ variations (100 days). As organic enrichment ($\alpha$) and thickness increase, the model estimates a longer

relaxation time with a maximum time span of 105 days.

Similar variation of relaxation time for DIC was simulated for different $\alpha$ and sediment deposit thickness. However, unlike $SO_4^{2-}$, relaxation time for DIC varies smoothly across the range of $\alpha$ and thickness combinations with relatively constant relaxation time (100 days) at low thickness and $\alpha$ combinations. The relaxation time increased exponentially as sediment





deposit thickness and labile OC concentration increased ($\alpha$), with maximum recovery time (171 days / 6 months) simulated at
the extremes of both combinations (Fig. 9).

## *4*      **Discussion**

In highly dynamic coastal ecosystems, such as RiOmar systems, driven by seasonal variability and meteorologically extreme
events, the response of early diagenetic processes to time varying deposition of organic matter is generally non-stationary (Tesi
et al., 2012). While dynamic equilibrium as a steady state condition may be reasonable in the case of seasonal variability, such
an assumption may fail in cases of instantaneously event-driven deposition. An intermittent supply of sediment and OC, like
those presented here, can cause a change in the system's properties on a short or long-term basis. Previous works have
highlighted excursions in sediment redox boundary (Katsev et al., 2006), flux of solutes at the sediment-water interface
(Rabouille and Gaillard, 1990) as well as modification of other system properties due to depositional flux of organic matter.
Thus, the premise of steady state conditions in early diagenetic processes which often depends on the temporal resolution of
the observation, might need revisiting especially in areas of episodic sedimentation (Wheatcroft, 1990; Tesi et al., 2012). Here,
we discuss the evolution and dynamics of a non-stationary sedimentary system following a singular perturbation.

### *4.1*      **Model representation and utility**

Non-steady state models are increasingly being applied in dynamic coastal environments, but they are still primarily based on
forcing from smooth varying boundaries that mimic seasonal forcing or long-term variability (Soetaert et al., 1996; Rabouille
et al., 2001a; Zindorf et al., 2021). Explicit consideration of abrupt changes in the upper boundary of the model caused by
events such as landslides, flash flooding, turbiditic transfer of materials on a continental slope, and trawling is still relatively
uncaptured by these models (but see De Borger et al. (2021) for inclusion of erosion events). In this paper, we adapt OMEXDIA
(Soetaert et al., 1996), a well-known reaction transport model, to investigate the changes in the solid and liquid phases during
massive deposition and mixing event. Our efforts highlight the algorithm's utility in incorporating this process with minimal
numerical issues. The model represented the basic characteristics of the data derived from the November/December 2008 flood
event at Station A in the Rhone Delta's depocenter (Fig 4). The simulated flux was also in agreement with the estimate from
field data, as diffusive oxygen uptake (DOU) rate sampled 26 days after the event (8th December 2008) was $16.6 \pm 2.9$
mmol m$^{-2}$ d$^{-1}$ (Cathalot et al., 2010) while the estimate from the model was 18 mmol m$^{-2}$ d$^{-1}$. As the inclusion of such
discontinuity in PDE(s) presents numerical challenges in classic solvers, the implementation utilized by our model ensures
such difficulties are overcome. This is the result of improved development of solvers adapted to such problem (Soetaert et al.,
2010b). This difference in the approach employed here distinguishes ours from other published models (e.g Berg et al. (2003),
Velde et al. (2018)) with similar scientific motivation for time dependent simulation. Overall, the validation of the model
output with field observations lends some confidence in using the model in scenarios involving abrupt changes in boundary
conditions and investigating biogeochemical changes in the sediment as a result of such an event. This is despite the model





under-estimation of the amplitude of sulfate and DIC at depth which can be improve with better optimization of some parameters, especially those derived from previous study that might not be suited for such flooding regime or with better process resolution relating to these pathways. Nonetheless, there are advantages to this model especially in the case of episodic flood deposit event, where only a snapshot of data is available at any given time. Modelling tools capable of simulating this event with high fidelity can provide continuous information of the system state and help fill in data gaps needed to understand

the sediment's response on different timescales.

### *4.2*    **Role of end-member flood input OM in the diagenetic relaxation dynamics**

Flooding events can transport large amounts of material through the river to transitional coastal environments such as deltas and estuaries. River floods can account for up to 80% of terrigenous particle inputs (Antonelli et al., 2008; Zebracki et al., 2015), and they can have a significant impact on geomorphology (Meybeck et al., 2007), ecosystem response, and

biogeochemical cycles (Mermex Group., 2011). If the source materials have a different organic matter composition (Dezzeo et al., 2000; Cathalot et al., 2013), the rapid deposition of these flood materials can alter diagenetic reactions and resulting fluxes.

Furthermore, the relaxation timescale associated with the sediment recovery following this external perturbation can be important in term of the process affecting the biogeochemistry of solid and solutes species. With a series of numerical

experiments ranging in between two end-members of the input spectrum for flood events such as those in the Rhone prodelta (Pastor et al., 2018), our study revealed contrasting sedimentary responses as well as associated typical time scales at which porewater profiles relax back to undisturbed state. Using a simple metric for estimating relaxation timescale of the pertubation, our calculations for the first end-member scenario (EM1) show that the upper bound of the timescale of relaxation for oxygen is $5 \pm 3$ days, whereas it was approximately $2 \pm 2$ days for the second end-member scenario (EM2). This reflects the property

of oxygen, which quickly approaches a steady state situation after an event (Aller, 1998). This viewpoint is supported by an ex situ controlled laboratory setup. In their studies, Chaillou et al. (2007) demonstrated that after gravity levelled sediment was introduced, oxygen consumption quickly recovered to its first-day level, with a sharp response time of 50 minutes and gradual shoaling of OPD within five days. We conclude that the tiny difference in oxygen relaxation and diagenetic response between the two scenarios can be attributed to the slow kinetic degradability of the refractory carbon deposited in the first scenario

versus the labile nature of the deposit in the second scenario. This kinetically driven OM degradation has been extensively studied and provides the basis for the reactive continuum in early diagenesis models (Middelburg, 1989; Jørgensen and Revsbech, 1985; Burdige, 1991).

Other terminal electron acceptors (TEAs) such as $SO_4^{2-}$, relax toward natural variation over a longer timescale than oxygen. For EM1, our simulation predicts a sulfate relaxation time of 117 days with a 95 % confidence interval (CI) estimate between

92 days (lower CI) and 139 days (upper CI) days) while in the case of EM2, we estimate a sulfate relaxation time of 91 days with comparatively low temporal variability (lower CI - 80 and upper CI - 103 days). This difference in relaxation time is


caused by the differences in sediment characteristics and how their mineralization occurs over the sediment layer. In the first scenario, organic-rich sediment is buried by less reactive new material. The buried sulfate fraction is reduced faster than in the new layer above and controls the short-term recovery. As the buried carbon stock depletes and the physical imprint of the flood 570 deposition fades, the profile begins to revert to its pre-flood shape. The post-flood evolution for the second scenario (EM2), on the other hand, differs in that the OM is consumed in the classical manner, with decreasing sulfate consumption with depth, caused by top-down control of the OM flux that adds OM to the sediment surface.

Such a long-time lapse for the recovery of an element with a complex pathway, such as $SO_4^{2-}$, has been reported in the literature (Anschutz et al., 2002; Stumm and Morgan, 2012; Chaillou et al., 2007). Similarly, estimates from our simulation for each 575 end-member scenario indicate that mineralization products such as DIC have a longer relaxation time. This is especially true for the first scenario as opposed to the second, with evidence of slow convergence at depth within the simulation time scale for the first scenario. We estimate that DIC will recover to its pre-deposition state in 5 months for EM1 and in a comparatively shorter time for EM2 (3 months). This lag in DIC recovery could be attributed to the fact that its post-flood dynamics is governed by the slow decaying detrital material that contribute to the already buried refractory carbon. This long term quasi-580 static behaviour of the porewater concentration despite such dynamic introduction of flood input can be understood by introduction the concept of a "*biogeochemical attractor*" effect - a similar analogy to the Lorenz attractor (Lorenz, 1963). This idea derived from the mathematical theory that describes chaos in the real world (Strogatz, 2018; Ghil, 2019). The existence of a "biogeochemical attractor" may explain why multiple temporal data sets in the Rhone River prodelta show a similar diagenetic signature from spring to summer (Rassmann et al., 2016; Dumoulin et al., 2018). Our timescale analysis estimates 585 that such rapid system restoration is indeed plausible and of the correct order of magnitude, based on the range of uncertainty reported here.

In addition, our calculations show that the time scale of return to the previous "pre-flood" profile is bracketed by the range of recovery due to purely molecular diffusion, putting an upper bound on our estimate. For example, using the Einstein's approximation, a species such as oxygen with a sediment diffusion coefficient (Ds) of 1.52 $cm^2 d^{-1}$ takes approximately 300 590 days to be transported solely by diffusion through a 30 cm sediment column and approximately 30 days for a 10 cm sediment column. Similar scaling argument could be made for species such as $SO_4^{2-}$ (Ds = 0.86 $cm^2 d^{-1}$) with > 500 days to be transported through 30 cm and ≈ 60 days for 10 cm. Because our estimates are less than these values, it suggests that processes other than diffusion (Thickness effect) may contribute to relaxation control. It emphasizes the importance of biogeochemistry (OM kinetic) in modulating the response after the event. Besides that, any long-term recovery timescale is governed by the 595 solid deposited. In comparison to the time scale of relaxation roughly estimated from field data (Cathalot et al., 2010), our estimate shows the right order of magnitude.

The relaxation time may also vary depending on the diagenetic interaction, and the characteristics of the organic matter available for degradation. This difference in characteristics was partially imposed in our study by assuming variations of $\alpha$ in the new deposit. The empirical observation of sediment characteristics associated with flood input dictates this parametric





turning to match the TOC characteristics (Pastor et al., 2018; Deflandre et al., 2002; Mucci and Edenborn, 1992; Tesi et al., 2012; Bourgeois et al., 2011). However, more data from the field and laboratory experiments that resolve the OM composition of flood deposits are required to constrain the choice of this numerical parameter.

### *4.3*   **Control of relaxation time by sediment deposit properties**

With the sensitivity analysis, we further explore the variation of relaxation timescale under variation of the thickness of layer
and enrichment factor of input material given by $\alpha$ in our model. The model's sensitivity analysis reveals that the thickness and concentration of the reactive fraction of TOC control the relaxation time across a wide range of deposited sediment perturbation characteristics (Fig. 9).

In terms of the recovery time as a function of the availability of labile OC, our results revealed a contrasting pattern for oxygen and sulfate. Several factors related to how different oxidants react with sediment matrix disturbances can explain these
differences:

- With oxygen that has a high molecular diffusion coefficient, variations in relaxation time depend on the levels of labile OC, with thin sediments containing a high level of labile OC showing a shorter recovery time than thicker sediments with a low OC content. This pattern can be attributed to the higher relative importance of oxygen consumption in OM poor sediment relative to the OM rich sediment.

- For low thickness deposits, sulfate and DIC relaxation times were more or less constant. However, a longer relaxation time was simulated for larger deposits and higher labile OC. This can be attributed to the increased distance required for solutes to migrate back after the event. This is clearly the case for sediment thicknesses greater than 14 cm. Such two-way dynamics could be explained by the fact that biological reworking and physical mixing within the surface mixing layer (SML) can improve OC degradation by promoting the replenishment of electron acceptors (i.e., oxygen,
sulfate, nitrate, and metal oxyhydroxides) (Aller and Aller, 1992; Aller, 2004); resulting in a shorter recovery time for the porewater profile to reorganize when perturbed below this depth.

This critical depth could also be the distance horizon at which the slow diffusion of the profile when retracting back to its pre-flood profile becomes an important factor in controlling the relaxation timescale. This is especially true for DIC, where the connection is more obvious. It has been proposed that when flood deposits extend beyond the sediment
bio-mixing depth, the relaxation time for the constituent species is determined by the concentration gradient between the historical and newly deposited layers (Wheatcroft, 1990). In our sensitivity analysis, higher $\alpha$ corresponds to higher $C_{org}^{fast}$ concentrations at depth, resulting in a case of enhanced OC degradation (both at the surface and within the sediment matrix). This depletes electron acceptors such as sulfate, which are required for OM mineralization at this depth. The slow diffusion across the displaced distance, on the other hand, cannot quickly compensate for its




demands, which may explain the longer relaxation time. In other words, a higher concentration of OC in a region where all oxidants are nearly consumed results in a profile that takes a relatively longer time to recover to its previous state due to the constraints imposed by oxidant availability. This viewpoint is consistent with previous research from the Rhone prodelta area, where a minimum transport distance of 20 cm is suggested for efficient connection with the SWI; above which several processes are decoupled (Rassmann et al., 2020) as well as other eutrophic systems, where

evidence of large accumulation of organic matter in subsurface sediments serves as a constraint on system restoration (Mayer, 1994; Pusceddu et al., 2009). Indeed, more observational and experimental studies are needed to better understand these processes.

### *4.4*    **Relaxation time metric: Limitation and perspective**

While one main focus of this study is on providing a quantitative estimate of relaxation time, the difficulty of objectively

defining what "*relaxation*" means necessitates some commentary. This difficulty is not unique to marine biogeochemistry, as accurate quantification of recovery time is an open research question in other fields. In the context of a sedimentary system, Wheatcroft (1990) proposed that determining "dissipation time" (analogous to our "relaxation time") can be subjective when it comes to signal preservation after sediment event layer deposition. The difficulties are exacerbated by previous work on episodic pulse on sediment biogeochemistry (Rabouille and Gaillard, 1990), in which two metrics for estimating relaxation

timescale for silica were proposed. Outside of benthic early diagenesis, Kittel et al. (2017) proposed two generic metrics for systems with well-defined asymptotic properties that can be applied to a distance function from a given target (subject to certain mathematical assumptions). Because porewater profiles are inherently nonlinear, and biogeochemical pathways in sediment are tightly coupled, the mathematical suggestion of asymposticity using such a distance metric for an evolving profile converging toward the "*target*" proposed in that paper is frequently not met. This is the case for our investigation. Overall,

while we provide a first-order approximation of relaxation time following perturbation for some model state variables, these studies highlight also some of the challenges associated with defining the timescale at which a signal can be validly assumed to have returned to its prior state. However, our method allows a full discussion of relaxation times for the main biogeochemical pathways.

### *5*    **Conclusion**

The need to comprehend extreme events and their relationship to marine biogeochemistry prompted the development of novel methods for diagnosing flood-driven organic matter pulses in coastal environments. In this paper, we propose a new model for characterizing flood deposition events and the biogeochemical changes that result from them. This type of event can have an impact on the benthic communities and the response of the whole ecosystem (Smith et al., 2018; Bissett et al., 2007; Gooday, 2002). Our modelling study shows that the post-depositional sediment response varies depending on the input characteristics

of the layer deposit. For instance, we tested the combined effect of enrichment of labile organic carbon and deposit thickness.





on space-time distribution and relaxation time of key dissolved species (oxygen, sulfate, DIC). This integral timescale of relaxation is constrained by the intrinsic properties of the solutes (diffusion) as well as the characteristics of the flood input (thickness and concentration of labile organic carbon). In essence, the findings from this study highlight the importance of the quantity and quality of organic carbon in modulating the sediment response following such a singular perturbation, as well as

the role of flood events with heterogeneous quantitative contributions in the coastal ocean.

## 5    Appendix

### A1    Biogeochemical reaction

The full model equation explained in section 2.3.2 is described fully below. Organic matter is composed of three fractions: fast degradable organic matter, slow degradable organic matter and refractory organic matter. Given the longtime scale for the

degradability of the refractory OM, it is parameterized using $TOC_{ref}$ as the asymptotic value. For the two other fractions, five mineralization pathways are included: aerobic respiration (AP), denitrification (DE), dissimilatory iron reduction (DIR), sulfate reduction and methanogenesis (MG).

Degradation of organic matter:

$$
\begin{aligned}
Cprod \quad &= \left( rFast \times C_{org}^{fast} + rSlow \times C_{org}^{fast} \right) \times \frac{(1-\phi)}{\phi} \\
Nprod \quad &= \left( rFast \times C_{org}^{fast} \times NCratio_{C_{fast}} + rSlow \times C_{org}^{fast} \times NCratio_{C_{slow}} \right) \times \frac{(1-\phi)}{\phi}
\end{aligned}
\qquad (A1.2)
$$

Limitation terms:

The limitation of a mineralization pathway by the availability of the oxidant is modeled by a Mond-type hyperbolic limitation function with inhibition of a pathway represented by a reciprocal hyperbolic function.

$$
\begin{aligned}
Oxicminlim \quad &= \frac{O_2}{O_2 + k_{O_2}} \times \frac{1}{lim} \\
Denitrificlim \quad &= \frac{NO_3}{NO_3 + k_{NO_3}} \times \left( 1 - \frac{O_2}{O_2 + k_{inO_2 den}} \right) \times \frac{1}{lim} \\
Feredminlim \quad &= \frac{FeOH_3}{FeOH_3 + k_{FeOH_3}} \times \left( 1 - \frac{NO_3}{NO_3 + k_{inNO_3 ano}} \right) \times \left( 1 - \frac{O_2}{O_2 + k_{inO_2 ano}} \right) \times \frac{1}{lim} \\
BSRminlim \quad &= \frac{SO_4}{SO_4 + k_{SO_4}} \times \left( 1 - \frac{FeOH_3}{FeOH_3 + k_{inFeOH_3 ano}} \right) \times \left( 1 - \frac{NO_3}{NO_3 + k_{inNO_3}} \right) \times \left( 1 - \frac{O_2}{O_2 + k_{inO_2 ano}} \right) \times \frac{1}{lim} \\
Methminlim \quad &= \left( 1 - \frac{SO_4}{SO_4 + k_{inSO_4 ano}} \right) \times \left( 1 - \frac{FeOH_3}{FeOH_3 + k_{inFeOH_3 ano}} \right) \times \left( 1 - \frac{NO_3}{NO_3 + k_{inNO_3 ano}} \right) \times \left( 1 - \frac{O_2}{O_2 + k_{inO_2 ano}} \right) \times \frac{1}{lim}
\end{aligned}
$$





$$lim = \frac{1}{Oxicminlim + Denitrificlim + Feredminlim + BSRminlim + Methminlim} \quad (A1.4)$$

Depth dependent kinetic reaction:

This limitation is used to reconstruct the vertical distribution of the successive mineralization pathways with a rescaling term $lim$ to ensure that the sum of the individual pathway equal the total degradation rate.

$$
\begin{aligned}
Oxicmin &= Cprod \times Oxicminlim \times lim \\
Denitrific &= Cprod \times Denitrificlim \times lim \\
Feredmin &= Cprod \times Feredlim \times lim \quad (A1.5) \\
BSRmin &= Cprod \times BSRlim \times lim \\
Methmin &= Cprod \times Methminlim \times lim
\end{aligned}
$$

Secondary reaction:

The Re-oxidation of reduced substance and other secondary reactions are modelled with a first order reaction term.

$$
\begin{aligned}
Nitri &= R_{nit} \times NH_4 \times \frac{O_2}{(O_2 + ks_{nitri})} && \text{(Nitrification)} \\
Feoxid &= R_{FeOH_3} \times Fe \times O_2 && \text{(Iron oxidation)} \\
H2Soxid &= R_{H_2S} \times H_2S \times O_2 && \text{(sulfide oxidation)} \\
CH4oxid &= R_{CH_4} \times CH_4 \times O_2 && \text{(Methane oxidation)} \quad (A1.6) \\
AOM &= R_{AOM} \times CH_4 \times SO_4 && \text{(Anaerobic oxidation of methane} \\
FeSprod &= R_{FeSprod} \times Fe \times H_2S && \text{(FeS production)} \\
DICprodCH4 &= -0.5 \times Methmin + CH4oxid + AOM && \text{(DIC production from methane)}
\end{aligned}
$$

Removal of sulfide via FeS production and oxidation with oxygen:

$$
\begin{aligned}
FeSprod &= R_{FeSprod} \times Fe \times H_2S && \text{(FeS production)} \\
H_2S_{oxid} &= rH_2S_{oxid} \times O_2 \times H_2S && \text{(Sulfide oxidation)}
\end{aligned} \quad (A1.7)
$$

Rate of change in state variable:





$$\frac{\partial C_{org}^{fast}}{\partial t} = transport + rFast \times C_{org}^{fast}$$

$$\frac{\partial C_{org}^{slow}}{\partial t} = transport + sFast \times C_{org}^{slow}$$

$$\frac{\partial O_2}{\partial t} = transport - Oxicmin - 1.5Nitri - 0.25FeOxid - 2H2Soxid - 2CH4oxid$$

$$\frac{\partial NH_3}{\partial t} = transport + \frac{(Nprod - Nitri)}{(1 + NH3ads)}$$

$$\frac{\partial NO_3}{\partial t} = transport - 0.8Denitrific + Nitri$$


$$\frac{\partial CH_4}{\partial t} = transport - DICprodCH4$$   $(A1.8)$

$$\frac{\partial DIC}{\partial t} = transport + Cprod + DICprodCH_4$$

$$\frac{\partial Fe}{\partial t} = transport + 4 \times Feredmin - Feoxid - FeSprod$$

$$\frac{\partial FeOH_3}{\partial t} = transport + (Feoxid - 4Feredmin) \times \frac{\phi}{(1 - \phi)}$$

$$\frac{\partial H_2S}{\partial t} = transport + 0.5BSRmin - H2Soxid - FeSprod + AOM$$

$$\frac{\partial SO_4}{\partial t} = transport - 0.5BSRmin + H2Soxid - AOM$$

**Author contributions.**

All authors contributed to the paper in several capacity. The project was supervised by CR and EV. SN, CR, EV conceptualized the method design, result interpretation, and assist in the initial draft of the paper. Model development was jointly design by

KS and SN. LP and BL provided insight on the data used in the model.

**Code availability.**

As a whole, the model is bundled as a R package for easy accessibility and can be downloaded from R-forge (**https://r-forge.r-project.org/R/?group_id=2422**). The most recent version of the model, as well as its evolution, can be found on the project development page (https://r-forge.r-project.org/projects/diagenesis/) with subsequent expected release in CRAN. Full R

vignette illustrating the capabilities of the model can be found on the model doc folder. The version used to produce the results used in this paper is archived on Zenodo (**https://doi.org/10.5281/zenodo.6369288**), along with the input data and scripts to recreate the simulation presented in this paper. FESDIA users should cite both this publication and the relevant Zenodo reference.

**Data availability**





The data and paper used to evaluate the model (Pastor et al., 2018) can be found in the Zenodo link. Users of the data should cite Pastor et al., 2018 and Ait-Ballagh et al 2021.

**Competing interests**

The contact author has declared that neither they nor their co-authors have any competing interests.

**Financial support**

This research has been supported by grant from INSU EC2C0 DELTARHONE and PhD grant from Ecolé doctorale des science de l'environment, Ile de France (SEIF) N° 129.

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
