# Peer review of "FESDIA (v1.0): Exploring temporal variations of sediment biogeochemistry under the influence of flood events using numerical modelling"

_Geoscientific Model Development, 2022_

## Author Comment (AC2)

**COMMENT AND RESPONSE FOR REFEREE 1**

We thank the reviewer for carefully reading our work and taking time to comment on some of strengths and perceived weaknesses of the paper. Our responses are represented in the red text in reply to the reviewer comment in black.

(1) Model description It would be helpful if the authors can provide a short overview of model development relative to the previous and already published works/models. Current manuscript referred to these works, but what is exactly new is kind of obscure. Also, brief summary of the capacity/features of previous models from which the current model has been developed will be useful, including programming language, governing equations, and algorithms utilized for numerical solutions (finite difference/volume/element method for equation differencing and Newton iteration if adopted for solution seeking etc.). The current manuscript provides some of above information, but some information is still missing. And it is not 100% clear which part is new to the current model and which part is not new.

The goal of this study is (1) to better understand the impact of episodic events on biogeochemical processes following deposition, and (2) estimate relaxation timescale. In order, to accomplish this, we developed an early diagenetic model called FESDIA based on early OMEXDIA code achievements. The ability to explicitly simulate non-steady early diagenetic processes in systems subject to perturbation events such as massive flood or storm deposition is a novel contribution of FESDIA to early diagenetic models. The novelty introduced by FESDIA compared to OMEXDIA are:

- An explicit description of the anoxic diagenesis beyond the nitrogen cycle including (i) Iron and Sulfur dynamics, (ii) methane production and consumption. In comparison OMEXDIA has a single state variable (ODU: oxygen demand unit) to describe reduced species. This addition required substantial rewrite of the code. This inclusion of the other primary pathways involving Fe, S, $CH_4$, and potentially Mn species, makes the model suitable for studies in region where organic matter input trigger of anoxic diagenetic processes.

- possibility to include sediment perturbation events such abrupt deposition of sediment. This is a crucial addition that makes the model suited for its intended application.

In this paper, we only discuss part of the FESDIA model concern with implementation of a perturbation event as it relates to some biogeochemical indicators. The model is implemented in Fortran (for speed) and linked to R (for flexibility).

(2) Definition of relaxation time Eq. 22 describes the change rate of species concentration within sediment profile but does not necessarily define deviation from the pre-flood profile. Also, current manuscript lacks figures that directly compare the solute profile development relative to the preevent profile. I think Figs. 4-8 need to be improved so that profile development relative to the pre-event profile is more visually obvious.

This relaxation timescale calculation based on the disappearance of the perturbed signal (via successive profiles similarities) may differ from an approach in which the profile returns to a pre-defined "old profile". Because the exactness of pre-flood and post-flood profiles is difficult to quantify numerically (Wheatcroft, 1990), and since the return to the former is frequently driven by slow dynamics, the approach used here can provide a window of estimate for which a particular signal fades toward the background of a theoretically pre-perturbed signal.

The solute profile is emphasized relative to their post-flood evolution because we are interested not on the exact return to the preflood profile (which can be rather long-term or never be if the system moves to a new steady state). Instead, we aim to characterize the time taken for a quiescence of the perturbed signal following the event (a much more reliable/quantifiable entity).

(3) Description of model limitations and future development Relevant to the above point, but limitations of model should be discussed more. There are several simplifying assumptions in the model but its influences on e.g., model validation, comparison with observation and estimation for relaxation time are not discussed.

A new section to discuss the model limitation has been added to revised version.

For instance, the authors assume that burial rate/porosity does not change with the flooding, but it is not discussed whether this assumption is defendable or close to what we observe.

Indeed, some coastal sediment burial rates have been shown to vary seasonally (Soetaert et al., 1996; Boudreau, 1994). In the proximal zone of Rhône prodelta, about 75% of sediment deposition occurs generally during the flood (eg., 30 cm/d). Remaining 25% is distributed along the year with a low range daily constant rate (0.03 cm/d). As well, porosity vertical distribution is time independent i.e. only organic matter content and deposition height change in the new layer. In the case of the Rhône river outlet, porosity in the flood deposit will depend on the contributing zones in the watershed and nature of eroded material as well as on rainfall intensity and duration. Resulting porosity in the new layer is barely predictable and could range between 0.65 and 0.85 in the proximal zone of the prodelta (Grenz et al., 2003; Cathalot et al., 2010).

Model validation or comparison with the observed data is essentially based on solute profiles, which likely resulted from a good fit of TOC and may be achievable under different assumptions (those that allow changes of burial rate, porosity, bio-mixing and irrigation etc.). If this is the case, the relaxation time is likely quite different under different assumptions. While most of manuscript

discussed how relaxation time is calculated under the specific assumption adopted for this study, it is not discussed how the relaxation time is affected by adopted assumptions.

Daily burial rate, porosity, bio-mixing and irrigation coefficients were extracted from the modelling work published in Pastor et al. (2011) and Ait-Ballagh et al. (2021) where only bio-mixing is an adjusted parameter. Other have been estimated from direct measurements and from macrofauna field observation. Adjustment of bioturbation coefficients for different sampling/modelled stations in the Rhône prodelta showed that modelling outputs were not clearly sensitive to these parameters (Ait-Ballagh et al. 2021).

**--- SPECIFIC COMMENTS ---**

In model description, it may be better if you say what programming language you are using earlier on (even in abstract).

The model is implemented in Fortran (for calculation speed) and linked to R (for flexibility).

L134. Three OM fractions? Thought the authors are using two.

This sentence containing this statement have been clarified in the revised paper by rephrasing:

"*As a result, the model assumed solid phase organic carbon with two reactive fractions with different reactivities and C/N ratios (Westrich and Berner, 1984; Soetaert et al., 1996). The degradation of OM involvesa labile fraction mineralizing faster than the slow decaying carbon. During the timescales considered here, the refractory organic matter class is not reactive. To compare with the observation, we consider an asymptotic OC constant ($C_{ref}$) for the inert fraction that scales the model calculated TOC output to the observation (Pastor et al., 2011) (see section 2.2.8).*"

L138. Froelich et al. (1979) rather than Froelich (1988) according to Bethke et al. (2011, AJS 311, 183)?

This has been corrected in the revised text.

L166. What does 'a coupled reaction formulation' mean?

The sentence was modified to read as: "*The law of mass action mathematically describes these reactions, with their kinetic rate term influencing the reaction.*"

Eqs. 6, 7. It does not make any sense to use/define 'maximum rates' when one is not using Monod or Michaelis-Menten type of equation.

"maximum rates" and replaced with "rates"

Eq. 7. What is the definition of rH2Soxid?

Text formatting issue with Microsoft when we translated our text from latex to Word. We fixed that in the updated text.

Eqs. 8, 9. What is the assumption behind the formulation of these equations? For instance, how do you obtain Eq. 9 for aqueous NH4+ with accounting for adsorbed NH4+? Can you track NH4+ adsorbed onto solid species along with OM and other solids or do you have to simplify that adsorption is depth-independent and/or time-independent? This can be important if solid materials with unoccupied exchange sites are flooded to sediment depocenter in a short period of time. If such case is possible, one would expect a large sink of NH4+ to the exchange sites? Related to this, do you model PO4 adsorption onto Fe hydroxides or you do not have to do this?

In regard to this comment, the sentence was modified to:

"*With special consideration of ammonium adsorption to sediment particles, the governing equation is given by:*

$$\frac{\partial \phi C}{\partial t} = \frac{-\partial}{\partial z}\left[\frac{-\phi \times D_{sed}}{(1 + k_{ads})} \times \frac{\partial C}{\partial z} + w_{\infty} \times \phi_{\infty} \times C\right] + \sum \frac{\phi \times REAC}{(1 + k_{ads})} \quad (9)$$

*where we assumed that the immobilization of $NH_4^+$ is in instantaneous, local equilibrium (i.e. any changes caused by the slow $NH_4^+$ removal process results in an immediate adjustment of the $NH_4^+$ equilibrium; so, can be modelled with a simple chemical species) and $k_{ads}$ is the adsorption coefficient. The inclusion of this formulation for the diffusion and reaction term has the effect of slowing down ammonium migration in sediment. Derivation of this formulation is given in Berner, 1980; Soetaert and Herman, 2009.* "

It is possible to calculated and tracked the amount of absorbed $NH_4$ as a modeled diagnostic which is part of the output of model.

The code has possibility to incorporate this Fe-P adsorption but that is not the scope of the paper which is focus on the main early diagenetic pathways affected by the flood deposition.

L205. Does porosity 'decay'?

The "porosity decay coefficient with depth" has been replaced with "depth constant for decrease of porosity with depth" (see Rabouille and Gaillard, 1991a)"

L209. According to Eq. 10, the authors seem to assume intraphase biodiffusion (Meysman et al., 2005, GCA 69, 3601). The statement here mentioning an interphase biodiffusion is inconsistent with Eq. 10.

It has been corrected with "*In this work, this bioturbation flux is assumed to be intraphase, with porosity $\phi(z)$ remaining constant over time.*"

Eq. 14. Irrigation term is not found in the governing equation. Is it included as a reaction term?

Irrigation is included in the REAC term shown in Eq 10 and to that effect a sentence has been added in L207 reading – "*This term includes both biological and chemical reaction within the sediment column as well as non-local bio-irrigation transport term (see next section).*"

Section 2.2.5 & Section 2.2.6. More details are desirable as adding grid for implementing a deposition event must be an important addition to the previous modeling framework. For instance, how you define Zpert, e.g., number of grids and their geometry etc. Some examples, not only schematics may also be useful.

Definition of the Zpert (as the depth of deposition of flood derived sediment layer) were briefly given in the main text. This scalar value is derived from the specified number of grid layer to perturb (Npert). The depth integration of the output from an irregular grid-based generation routine of the ReacTran R packages (which implements many grid types used in early diagenesis modelling) for a number of grid points (Npert) results to Zpert. An example can be found in Figure S1 of the supplement text.

This explanation is added in the section 2.2.5 in the revised text.

L276. 'maximum in the spring and minimum in fall and winter'. This line does not make sense to me.

This sentence has been rephrased with: "*In the latter case, this carbon flux varies over the annual carbon flux ($\overline{flux_{org}}$) in the region in question.*"

Eq. 21. What is the units of TOC? Also, how do you derive Eq. 21? Is this simply analytical solution of the governing equation? In any case, it would be helpful if the authors can provide the procedure to obtain Eq. 21 somewhere.

The unit is percent dry wet. A preamble like this has been added to the revised text:

"*TOC (in % dw) is estimated from both carbon fractions ($C_{org}^{fast}$ and $C_{org}^{slow}$) assuming a sediment density ($\rho$) of 2.5 $g\ cm^{-3}$ and conversion from the model unit for detrital carbon fraction of $mmol\ m^{-3}$ to unit percent mass. The TOC is not analytical solution to the general governing equation. It is diagnosed from both the fast and slow decaying detritus component of the OM which are calculated from the governing equation that is solved numerically*"

A background refractory concentration Cref is added to Eq 21 to account for the fraction which are not calculated within timescale simulated by the model.

L321. 'were utilizes the R programming language'. Correct English?

*This has been corrected to: "The R programming language is used in the preprocessing routine for model grid generation (Soetaert and Meysman, 2012), porewater chemistry parameter (Soetaert et al., 2010a), steady state calculation (Soetaert, 2014), and time integration (Soetaert et al., 2010b)".*

Section 2.2.9. It would be helpful if the authors can make a short description of what part of "deSolve package" they used, not only directing the reader to the R-forge webpage. More specifically, how the authors numerically solve the governing equations, apart from "method-of-lines" methods? Use of any finite difference/volume/element method?

*This is also a featured requested by the reviewer 2. We added in section 2.2.9 a new paragraph:*

*"Because the procedure is based on OMEXDIA, complete details of the derivation can be found in that paper and referenced therein (Soetaert et al., 1996). Here we recap the mathematical formulation of the method-of-lines (MOL) algorithm used by FESDIA. Direct differencing of (8) - (10) results to:*

$$\frac{\partial C_i}{\partial t} = \frac{\Phi_{i,i+1} D_{\Phi_{i,i+1}}(C_{i+1} - C_i)}{\Phi_i \Delta x_{i,i+1} \Delta x_i} -$$
$$w_\infty \Phi_\infty \frac{\alpha_{i,i+1} C_i + (1 - \alpha_{i,i+1}) C_{i+1}}{\Phi_i \Delta x_i} -$$
$$\frac{\Phi_{i-1,i} D_{\Phi_{i-1,i}}(C_i - C_{i-1})}{\Phi_i \Delta x_{i-1,i} \Delta x_i} +$$
$$w_\infty \Phi_\infty \frac{\alpha_{i-1,i} C_{i-1} + (1 - \alpha_{i-1,i}) C_i}{\Phi_i \Delta x_i} \qquad (22)$$

*for a generic tracer $C$ with a phase properties index $\Phi$ and $D_\Phi$ denoting porosity and dispersive mixing term respectively for solid or liquid. This equation is calculated such that the variables and parameters are defined both at the centre of each layer $x_i$ and at the interface between layers ($x_{i,i+1}, x_{i,i+1}$). The position at the centre of the grid is then given as $x_i = \frac{x_{i-1,i} + x_{i+1,i}}{2}$. $\Delta x_i$ represents the thickness of the i-layer and $\Delta x_{i,i+1}$ is the distance between two consecutive grid layers. A Fiadeiro scheme (Fiadeiro and Veronis, 1977) based on the model's Peclet number (a dimensionless ratio expressing the relative importance of advective over dispersive processes) is used to provide a weighted difference of the transport terms in order to reduce numerical dispersion.*

How is the time-integration of governing equations made (time-implicitly or -explicitly)?

*"...Equations (8)-(10) implemented as Eq. (22) is integrated in time using an implicit solver, called lsodes, that is part of the ODEPACK solvers (Hindmarsh, 1983). This solver uses a backward differentiation method (BDF); it has an adaptive time step, and is designed for solving systems of ordinary differential equations where the Jacobian matrix has an arbitrary sparse structure."*

L331. What is a "slow" stationary state?

The word and its surrounding sentence have been rewritten as follows:

"*We estimate the relaxation timescale by first calculating the absolute difference ($\varphi(t)$) between successive model output after the event, assuming that a slowly evolving state will eventually converge to the pre-perturbed state as time after the disturbance approaches infinity.*"

Eq. 22. Not quite sure this is a legit mathematical expression. phi(tau) < threshold is what I thought is consistent with what the authors described.

It was intended to imply that the relaxation estimate is based on calculating a threshold given by the median of the $\varphi(t)$. Eqn 22 have been updated to explicitly specify the subscript and avoid any misinterpretation:

$$\varphi(t) \quad = \frac{1}{N} \sum_{i=1}^{N} \|X_{t+1}^{i} - X_{t}^{i}\|$$

In the limit of time (t): $\qquad\qquad$ (23)

$$\tau(t) \quad \Rightarrow \varphi(t) \leq \varphi(t)_{threshold}$$

$$\text{where } \varphi(t)_{threshold} = \overline{\varphi(t)} \approx \text{seasonal background}$$

L334. "threshold (i.e given by the median over the entire time duration)." Do you mean that the run is finished when difference becomes less than the median value throughout the simulation and then tau is defined as the model time required for this?

Yes, Using the Eqn 22, the calculation terminates at the infimum (greater lower bound) time point when the curve encounters the threshold. That terminal time point is thus the $\tau(t)$ as given by the equation.

L340. I probably do not fully understand the ensemble of simulations here to estimate the uncertainty in tau. What parameter do you randomly re-sampled exactly? Median of the reference run through time? If so, the runs for determining the uncertainty in tau is conducted until (randomly-chosen) prescribed median is crossed? But this does not necessitate re-running of the model as the boundary conditions are not changed?

This section has been rewritten to and the pertinent parameter being varied is highlighted in bold:

"*In this case, we employ a modified bootstrapping technique to estimate the uncertainty in the relaxation timescale by resampling on the **cutoff point** introduced in Eq. 23 (i.e. median, $\overline{\varphi(t)}$ of a given reference simulation). This calculation takes advantage of the fact that the timeseries will be dominated by the **slowly varying seasonal cycle over a long time period away from the point of perturbation**, with the influence of the perturbation fading to the background. **The variation of this reference timeseries over time reflects the uncertainty in this median threshold point.** This variance, along with the reference cutoff value, can be used to generate n random perturbations varying about the normative threshold value. We can proceed to create a histogram of the replicate threshold(s) distribution. The histogram of*

*this distribution is depicted schematically in the left margin of Fig. 3. The relaxation time in each realization of the threshold is calculated ($\hat{\tau}_\iota$). The median absolute deviation from this ensemble of relaxation times is then used to calculate the level of uncertainty in the statistics of interest (timescale of relaxation - ($\hat{\tau}$)). Figure 3 depicts this concept schematically.*

*It should be noted that this method eliminates the need to rerun the deterministic model for each iteration, reducing the computational burden of this technique.*"

L383. "a thickness scale of 1 cm to 30 cm in 5 cm increments". This line does not make sense to me. What exactly did you use for thickness in sensitivity analysis?

This sentence has been changed to include the full value in the revised text.: "*...a thickness variation ranging from 1cm to 30 cm. A 5 cm increment was used for the sensitivity analysis*"

 Section 2.2.11.2. If the tested values are not too many, it would be better to list exact values you used for sensitivity analysis.

The revised text includes the tested values utilized for sensitivity analysis.

L507. Please specify what "RiOmar" stands for.

It is stand for river-dominated ocean margins. It is mentioned in the Introduction and discussion.

L621. above → below?

The preceding sentence have been changed to: *"...resulting in a shorter recovery time for the porewater profile to reorganize within the SML*"

This thereby explicitly specify where the shorter recovery time occurs.

**--- TECHNICAL COMMENTS ---**

Table 1. What does unequal mark on $Fe(OH)_3$ mean? Is this typo? At least notion should be consistent with that in main text.

It is a typo. It has been deleted in the revised paper.

L139. Eq. 3 → Eq. 2?

This has been changed in revised manuscript.

L165. Typo in the second line of Eq. 5.

The typo has been corrected in revised manuscript.

L168. Right parenthesis in the last line of Eq. 6 is missing.

This have been fixed.

L212. Where → where?

This has been corrected in the revised text.

L214. specify à specified?

"Specify" has been replaced with "specified" in the revised manuscript.

L230. i.e à i.e.?

This is fixed in the revised manuscript.

L232. occur → occurs?

The word is corrected in the revised text.

L246. Figure. 2 à Fig. 2 or Figure 2?

This Figure labelling in the text has been made consistent.

L318. "method-on-lines" à "method-of-lines"?

Indeed, this have been corrected to "method-of-lines" in the revised manuscript.

L409. dissolved DIC à DIC?

This has been fixed and corrected.

L415. as thus → as follows?

This has been corrected in the revised manuscript.

L418. Table. 3 à Table 3

A "." have added to Table in the revised manuscript.

L433. "Solid" should not be superscript

The superscript has been be removed in the revised text.

L434. "Solid" should not be superscript

Same as above.

L535. Improve → improved

It has been corrected accordingly in the revised manuscript.

L581. introduction à introduction of or introducing

It has been corrected accordingly.

L660. Thickness → thickness

Corrected.

L694. design → designed

Corrected.

**REFERENCES**

Alan C. Hindmarsh, ODEPACK, A Systematized Collection of ODE Solvers, in Scientific Computing, R. S. Stepleman et al. (Eds.), North-Holland, Amsterdam, pp. 55-64, 1983.

Boudreau, B. P.: Is burial velocity a master parameter for bioturbation? 58, 1243–1249, 1994.

Burdige, D.J. The biogeochemistry of manganese and iron reduction in marine sediments. Earth Sci. Rev., 35, pp. 249-284, 1993.

Cathalot, C., Rabouille, C., Pastor, L., Deflandre, B., Viollier, E., Buscail, R., Grémare, A., Treignier, C., and Pruski, A.: Temporal variability of carbon recycling in coastal sediments influenced by rivers: assessing the impact of flood inputs in the Rhône River prodelta, Biogeosciences, 7, 1187–1205, https://doi.org/10.5194/bg-7-1187-2010, 2010.

De Borger, E., Tiano, J., Braeckman, U., Rijnsdorp, A. D., and Soetaert, K.: Impact of bottom trawling on sediment biogeochemistry: a modelling approach, Biogeosciences, 18, 2539–2557, 2021.

Gaillard, J., Pauwels, H., and Michard, G.: Chemical diagenesis in coastal marine-sediments, 12, 175–187, 1989.

Grenz, C., Denis, L., Boucher, G., Chauvaud, L., Clavier, J., Fichez, R., and Pringault, O.: Spatial variability in sediment oxygen consumption under winter conditions in a lagoonal system in new caledonia (south pacific), 285, 33–47, 2003.

Krumins, V., Gehlen, M., Arndt, S., Van Cappellen, P., and Regnier, P.: Dissolved inorganic carbon and alkalinity fluxes from coastal marine sediments: Model estimates for different shelf environments and sensitivity to global change, 10, 371–398, 2013.

Rabouille, C. and Gaillard, J.-F.: Towards the EDGE: Early diagenetic global explanation. A model depicting the early diagenesis of organic matter, O2, NO3, Mn, and PO4, Geochimica et Cosmochimica Acta, 55, 2511–2525, 1991.

Rassmann, J., Eitel, E. M., Lansard, B., Cathalot, C., Brandily, C., Taillefert, M., and Rabouille, C.: Benthic alkalinity and dissolved inorganic carbon fluxes in the Rhône River prodelta generated by decoupled aerobic and anaerobic processes, Biogeosciences, 17, 13–33, https://doi.org/10.5194/bg-17-13-2020, 2020.

Rassmann, J., Lansard, B., Pozzato, L., and Rabouille, C.: Carbonate chemistry in sediment porewaters of the Rhône River delta driven by early diagenesis (northwestern Mediterranean), Biogeosciences, 13, 5379–5394, 2016.

Sciberras, M., Hiddink, J. G., Jennings, S., Szostek, C. L., Hughes, K. M., Kneafsey, B., Clarke, L. J., Ellis, N., Rijnsdorp, A. D., McConnaughey, R. A., et al.: Response of benthic fauna to experimental bottom fishing: A global meta-analysis, 19, 698–715, 2018.

Soetaer, K., Herman, P. M., and Middelburg, J. J.: Dynamic response of deep-sea sediments to seasonal variations: A model, Limnology and Oceanography, 41, 1651–1668, 1996.

Soetaert, K. and Meysman, F.: Reactive transport in aquatic ecosystems: Rapid model prototyping in the open source software R, Environmental Modelling & Software, 32, 49–60, 2012.

Soetaert, K., & Herman, P. A Practical Guide to Ecological Modelling: Using R as a Simulation Platform. Dordrecht: Springer, 2009. [doi:10.1007/978-1-4020-8624-3]

Soetaert, K., Petzoldt, T., and Meysman, F.: Marelac: Tools for aquatic sciences, 2010a.

Soetaert, K., Petzoldt, T., and Setzer, R. W.: Solving Differential Equations in R: Package deSolve, Journal of Statistical Software, 33, 1–25, https://doi.org/10.18637/jss.v033.i09, 2010b.

Soetaert, K.: Package rootSolve: roots, gradients and steady-states in R, Google Scholar, 2014.

Sulpis, O., Boudreau, B. P., Mucci, A., Jenkins, C., Trossman, D. S., Arbic, B. K., and Key, R. M.: Current $CaCO_3$ dissolution at the seafloor caused by anthropogenic $CO_2$, 115, 11700–11705, 2018.

Sundby B. And Silverberg N. Manganese fluxes in the benthic boundry layer. Limnol. Oceanogr. 30,372-38 1, 1985.

Tesi, T., Langone, L., Goñi, M., Wheatcroft, R., Miserocchi, S., and Bertotti, L.: Early diagenesis of recently deposited organic matter: A 9-yr time-series study of a flood deposit, Geochimica et Cosmochimica Acta, 83, 19–36, 2012.

Thamdrup, B., H. Fossing, and B. B. Joergensen. Manganese, iron, and sulfur cycling in a coastal marine sediment, Aarhus Bay, Denmark. Geochim. Cosmochim. Acta 58:5115–5129, 1994.

Turchyn, A. V., Bradbury, H. J., Walker, K., and Sun, X.: Controls on the precipitation of carbonate minerals within marine sediments, 9, 57, 2021.

Van Cappellen, P. and Wang, Y. Cycling of iron and manganese in surface sediments: a general theory for the coupled transport and reaction of Carbon, Oxygen, Nitrogen, Sulfur, Iron and Manganese. Amer. Jour. Sci. v. 296, 197-243, 1996.

Wheatcroft, R. A.: Preservation potential of sedimentary event layers, Geology, 18, 843–845, 1990.

---

## Author Comment (AC3)

**COMMENT AND RESPONSE FOR REFEREE 2**

We thank the reviewer for carefully reading our work and taking time to comment on some of strengths and perceived weaknesses of the paper. Our responses are represented in the red text in reply to the reviewer comment in black.

**OVERALL COMMENT**

Concerning the overall evaluation of the paper and its novelty, the reviewer acknowledge that novelty is present in this model by the addition of new diagenetic pathways and by the addition of the potential to calculate the effect of the sudden deposition of a new and thick layer of sediment with a different concentration of organic carbon (OC). In the following ways, FESDIA differs therefore from the OMEXDIA model by implementing:

- An explicit description of the anoxic diagenesis beyond the nitrogen cycle including (i) Iron and Sulfur dynamics, (ii) methane production and consumption. In comparison OMEXDIA has a single state variable (ODU: oxygen demand unit) to describe reduced species.

- possibility to include sediment perturbation events such abrupt deposition of sediment. This is a crucial addition that makes the model suited for its intended application.

In this paper, we only discuss part of the FESDIA model concern with implementation of a perturbation event as it relates to some biogeochemical indicators. The model is implemented in Fortran (for speed) and linked to R (for flexibility).

In regard to the paper structure, efforts have been made to improve on its clarity, and flow as a result of the reviewer feedback.

**GENERAL COMMENT**

Rassmann et al. (2016 https://doi.org/10.5194/bg-13-5379-2016, 2020 https://doi.org/10.5194/bg-17-13-2020) described Rhône river delta sediments rich in calcium carbonates, and reported signs of the various reactions associated with calcium carbonates occurring in those sediments. If the focus here is Rhône river delta sediments, why not including any calcium carbonate species in the current model?

Previous work on Rhône river delta sediments has revealed that the sediment contains calcium carbonates, which may have justified the inclusion of calcium carbonate in the model. However, Rassmann et al 2020 shown that the effect of calcite formation on the DIC concentrations is between can be at most below 15%. Carbonate system models have been discussed as computationally demanding (Boudreau, 1997; Hoffmann et al., 2008). In this paper and for this first version of the model, we intend to consider first order effects of flood deposition on the profiles, as well as to highlight the possibility of incorporating diagenesis changes in the sediment when including abrupt discontinuities. While other community tools, such as CANDI (Boudreau et al., 1996), MEDUSA (Munhoven, 2021) and the recently published RADI model (Sulpis et al., 2021), include the carbonate system, our aim here

is to provide tools with the ability to study perturbational dynamics, such as sudden depositional events. Efforts will undoubtedly be made in the future to integrate the complexity of the carbonate system into flood deposition routine like FESDIA.

Section 2.2.6: this seems to be the most important section in terms of model development, but it is also the part I had the hardest time to follow. There are a lot of new terms introduced here and they are not well defined. The second paragraph starts to explain how post-flood organic carbon contents are derived, then mentions the solutes, then goes back to post-flood organic carbon content with Eq.(15)… I suggest putting more effort clarifying this section, defining terms with precise and consistent words, in order not to confuse the reader.

We have modified this section and its organization as suggested. Without repeating the full text, the following snippet capture the changes we made with discussion of the deposition effect on the solids first and then the solutes:

*"...The event calculation was carried dynamically within the same time run. For the solid species, following the flux of organic carbon via the boundary condition (see section 2.2.7), the portion of organic carbon is split between the fast and slow decaying component using a proportionality constant ($pfast$) as in Ait Ballagh et al. (2021). $pfast$ varies from 0 to 1 and it is express in percentage of carbon flux deposited associated to either fraction (fast and slow). However, at the time when the event is prescribed, the integrated profile of the solid species $C_{org}^{fast}$ and $C_{org}^{slow}$ from previous time step, defined as ($t^-$), was used to create a virtual composite of the deposited layer. This integral calculation was performed over a specified sediment thickness ($Z_{pert}$), which corresponded to the vertical extent of the depositional event. This average concentration for the solid, which we define exclusively for the time of deposition as ($C_{org}^{flood}$) is scaled with an enrichment factor ($\alpha$) see below) and then nudged on top of the old layer which is supposed to be buried beneath after the event…*

*…The carbon enrichment factor ($\alpha$) in the model (confac in the model code) is introduced here in order to scale the deposited OC with those observed from field data. This helps in calibrating the deposited organic matter concentration ($C_{org}^{fast}$ and $C_{org}^{slow}$) in the new layer relative to the previous sediment fraction, simulating the wide range of TOC content observed in the field. For instance, when the newly deposited organic matter is similar to the former sediment topmost layer (average preflood layer concentration over an equivalent $Z_{pert}$ depth), an ($\alpha$) value of 1:1 is used. If the new material is lower in organic carbon content compared to what is near the sediment-water interface, then ($\alpha$) < 1, while if the newly deposited material is higher in carbon content than the sediment surface, ($\alpha$) > 1. This flexibility can be used to constrain the simulation to match the corresponding TOC profile from field observation. In modeling application, this parameter is generally specified by using different value for the magnitude of OC in each fraction depending on the empirical observation of the TOC data. This*

*quantity is therefore tunable and the upper bound of this parameter is dictated by the maximum TOC in the sediment sample…*

Specifically, what is the carbon enrichment factor (confac) exactly, and how does it differ from the proportionality constant (pfast)?

*"…It is important to note that this parameter differs from $pfast$. This OC flux partitioning by $pfast$ occurs regardless of the event and it is related to the carbon flux received at the boundary, but the carbon enrichment factor occurs only during the event. The Carbon enrichment factor ($\alpha$) can be viewed as a method of imposing a new initial condition only at the time of the event by using the integral concentration from the previous time. However, using the approach described here, all calculations can be done dynamically without stopping the model.*

*For the solutes ($O_2$, $NO_3^-$, $NH_4^+$, DIC, $SO_4^{2-}$), the bottom water concentration is imposed through the perturbed layer at the time of event by assuming this new layer is homogenously mixed…"*

What is $C_{org}^{flood}$ and how does it differ from TOC (both are present in Eq. (15)? Is confac tuned for each simulation or is it constant? Is pfast tuned for each simulation or is it constant?

*$C_{org}^{flood}$ is only a notational term defined for when the event occurs and helps to distinguish Corg (which we used consistently throughout the manuscript). It also differs from TOC. TOC, as defined in Eq 21, is not a modelled variable (not a state variable), rather, it is determined by the fast and slow degradable carbon. ($C_{org}^{fast}$ and $C_{org}^{slow}$) as well as considering for the refractory background carbon ($C_{ref}$).*

Is confac tuned for each simulation or is it constant?

*For any simulation, confac ($\alpha$) is changed to account for targeted variations in deposited carbon concentration during the flood. In the revised paper, A statement have been added to clearly indicate that this parameter is application specific and can be tailored to the data at hand.*

Is pfast tuned for each simulation or is it constant?

*For the majority of the applications presented in the paper, the pfast is constant, drawing heavily on Pastor et al., 2011's and previous modelling optimization of this parameter to data in the study region.*

It is stated L187-188 that "For dynamic simulation, w can change as a function of time but in most cases, we assumed a constant value." In which cases exactly was w changing? Changing w in all cases seem like a necessity given that the novelty of the model is to simulate events in which the flux of deposited material (thus w) is strongly changing with time. How can a constant w be appropriate to simulate a flood? w also changes with sediment depth, because of chemical reactions occurring within the sediment (see Munhoven, 2021 https://doi.org/10.5194/gmd-14-3603-2021). Can the authors either better justify their choice of a non-changing w or update that in the model simulations?

Given that the sedimentation rate in the Rhône prodelta can range from 10 cm to 41 cm/y and flood deposition can deliver about 30 cm in a few days (about 5 days), the burial rate in the non-flood period can be assumed to be constant (i.e., evenly distributed through the rest of the year). Because the deposition of the flood layer is treated separately by re-adjusting the depth profiles with the inclusion of the flood layer, the main variation of accumulation rate is already taken into account. Furthermore, because our model heavily borrows from OMEXDIA, where the flux of OC is decoupled from the advection rates (Soetaert et al., 1996a,b), we anticipate that the inclusion of time varying w will be marginal for the single flood application.

FESDIA model was designed in such a way that time-varying sedimentation is possible. The function FESDIAperturb() (discussed in section 2.2.9) has an argument (wForc) that can be given a functional time series or imposed as observational data for the sedimentation rates (w) if available. In the revised manuscript, we now highlight this possibility to potential readers/users.

w also changes with sediment depth, because of chemical reactions occurring within the sediment

Some modelling paper has alluded to this possibility (Munhoven 2021). However, the current version of FESDIA does not considered the vertical variation of w due to chemical reactions. Given that the flood input can be as high as 30 cm in some of this massive sediment depositional event, the change in advection rate due to chemical change is negligible in comparison. In addition, Munhoven 2021, noted that inclusion of this chemically induced change in advection rate required the formulation of a volumeless solid component which can results to "physically unrealistic transport" and so be "required only if necessary".

Section 2.2.7: in most O2 and pH microprofiles from the Rhône delta presented in Rassmann et al. (2016) we can see the influence of a diffusive boundary layer. Please discuss and justify the absence of diffusive boundary layer control on solutes as an upper boundary condition, or update the upper boundary condition accordingly to include this, as other models do in a simple manner (Boudreau et al., 1996 https://doi.org/10.1016/0098-3004(95)00115-8; Munhoven, 2021 https://doi.org/10.5194/gmd-14-3603-2021; Sulpis et al., 2022 https://doi.org/10.5194/gmd-15-2105-2022).

Including DBL in the model, particularly when it comes to $O_2$ and pH in the upper sediment can be necessary to calculate the benthic oxygen demand or to define accurately oxygen top boundary condition. However, a key focus of this paper was attempting to showcase approaches to establish an estimate for the relaxation timescale for dissolved chemical species whose zone of action occurs deeper in the sediment. Thus, the DBL zone was excluded from the model and its use-case scenarios as presented here. As evidenced by our findings, solutes such as $SO_4^{2-}$ and DIC are prominent examples where this assumption can be made. This simplification is consistent with previous research on the role of DBL in sediment flux and reactions (Boudreau & Jorgensen, 2001, Chapter 9), which proposed that the importance of DBL in controlling diagenetic processes and fluxes is determined by the relative ratio

of DBL thickness ($\delta_d$) to the depth of solute change in the sediment (L) (Fig 9.9 in Boudreau & Jorgensen, 2001). For species such as $SO_4^{2-}$ and DIC, where the depth of diagenesis change (L) is greater than the average thickness of DBL as found region close to the Rhône prodelta (0.12 cm, Sulpis, et al 2018), DBL will play only a minor role in controlling the flux and relaxation timescale.

However, by omitting DBL for solutes such as oxygen where $\delta_d/L \gg 1$, our model may overestimate the fluxes across the SWI. By varying the bottom water concentration accordingly to this DBL effect, calculated relaxation time for dissolved oxygen was longer of 1 day at most and do not really modify previous conclusion. Comments on the absence of DBL is now included in the discussion section where we now added a subsection on FESDIA limitations.

There are a lot of inconsistencies between number reported in the text and those in the tables (w, NC ratio, rslow, bottom boundary conditions). Please update and be consistent.

This is now corrected in the revised version.

**SPECIFIC COMMENTS:**

Shouldn't "Rhône" be spelled "Rhône", even in English language?

This is now corrected in the revised manuscript.

L21-24: Here the enrichment factor alpha is mentioned but not clearly defined. This is confusing. Please update.

See previous comment above. The revised manuscript has been updated accordingly.

**Introduction**

L35: The use of the acronym RiOmar is not really needed, since only used once after. In general, avoid unnecessary acronyms.

This acronym was used because other authors and potential readers might be familiar with this naming international convention. Acronym has been removed but "reference to river dominated ocean margins kept".

L36: Although more commonly used, POC is also an unnecessary acronym here, since only used once after.

POC has been removed.

L36-39: The sentence is unclear. "because it serves as a sink for particulate organic carbon and nutrients as well as an intense site of carbon and nutrient": what is the "it" referring to?

The sentence has been rephrased to: … *The fate of organic matter derived from riverine input to the sediment is of biogeochemical importance in coastal marine systems (Cai, 2011). This coastal environment serves as a sink for particulate organic carbon and nutrients, as well as an active site of carbon and nutrient remineralization (Burdige, 2005; McKee et al., 2004; Sundby, 2006)...*"

L40: I am not convinced that all the cited models have time-dependent capabilities, unlike several other, more recent models, published in this journal that explicitly do. Please update the list.

We are aware of some new modelling framework and tools published especially in this journal and the list will be updated.

(Lasaga and Holland, 1976; Rabouille and Gaillard, 1991; Boudreau, 1996; Soetaert et al., 1996; Rabouille et al., 2001a; Archer et al., 2002; Couture et al., 2010; Yakushev et al., 2017, Munhoven, et al, 2021, Sulpis et al 2022)

L43: "massive episodic events" could refer to lots of processes, please be more specific.

The sentence has been replaced with:

"*However, because of the scarcity of observations and their unpredictability, the role of massive deposit of sediment in these early diagenesis models has frequently been overlooked*".

L47-50: Sentence unclear. "Attempts to use mathematical models to understand perturbation-induced events on early diagenetic processes have resulted in a variety of approaches that incorporate this type of local phenomenon.": what is "this type of local phenomenon" referring to?

The sentence has been rephrased to:

"…*Attempts to use mathematical models to understand perturbation-induced events such as sudden erosion/resuspension event, bottom trawling, and turbidity driven sediment deposition on early diagenetic processes have resulted in a variety of approaches that incorporate this type of phenomenon…*"

The word "local" has been removed from the sentence in the revised manuscript.

L48-50: "As an example, previous research in deep-sea systems suggests that megafaunal perturbation can cause a 35% increase in silicic flux when compared to steady state estimates (Rabouille and Gaillard, 1990)" this is interesting but this level of precision seems unnecessary, what is the relevance for this study? Besides, what is a "silicic flux"? In which direction is the mentioned flux going?

This part has been removed.

L50: What is the "redox boundary"

According to Katsev et al. (2006), the redox boundary is defined as the depth zone beneath the sediment-water interface that separates the stability fields of the oxidized and reduced species of a given redox couple. In order to emphasize our use of the word in the text, we have provided a brief definition in the revised paper.

L52-53: What does the "redistribution of solid-phase manganese with multiple peaks" mean?

According to Katsev et al. (2006), the presence of temporal variation in the organic matter flux of Deep Artic sediment can result in a shift in the depth horizon of the so-called "redox boundary" (see definition above). This redox shift, along with the associated depth zone of oxidation and reduction, can influence where the deep manganese peak forms. That is what the paper (Katsev et al 2006) intends to convey, hence our use of the phrase "*redistribution of solid-phase manganese with multiple peaks*." To communicate this idea more clearly, the sentence has been restructured in the revised paper.

L62: "porewater species like oxygen (O2) can be restored after a few months": it is unclear. Do you mean that porewater concentrations can be restored to their pre-flood levels?

$O_2$ data indicate that their relaxation time is short (days to weeks at most), and that in the absence of another successive massive deposition over a short period of time, they can return to pre-flood levels. This is supported by data from multiple campaign observations (Rassmann et al., 2020).

The sentence has been rewritten "Vertical distribution of porewater species like oxygen ($O_2$), can be restored after a few days"

L66: what does "short-lived species" mean?

The short-lived species have been replaced with "species with short relaxation time"'.

L67: DIC is a component, not a species.

This has been corrected in revised manuscript.

**Materials and methods**

L93: "the organic matter delivered reflects the Rhône River inputs (Lansard et al., 2008; Cathalot et al., 2013)", in terms of what? Composition? Reactivity? Fig.1: I assume that the dashed and solid gray linings shown on the map depict bathymetry; it would be useful to precise it in a caption/legend

Organic matter delivered in this context is defined by its "composition." The Rhône prodelta, acting as a depocenter, can transport materials of various source compositions from the terrestrial domain (Pastor et al., 2018). However, the reactivity of the composite deposited sediment is not well defined in this area (we provided a comment on this in the discussion L599-L602). In the revised paper we will endeavour to emphasize the "compositional" distinction.

The caption/legend for the map will be included in the revised text.

L107: what does "mode of behavior" mean?

It has been deleted and replace with: "their evolution following the event…"

L140: how exactly do "the reactivities decrease with depth" in the present model? From Table S1, it seems that the reactivities are constant.

The proportion of reactive carbon to refractory carbon decreases with depth and reaction rate changes accordingly. This sentence has been rewritten in the next version.

L140: The sentence formulation is awkward: it is the degradation that would "cease", not its rate. Saying this also slightly exaggerated, degradation rates become indeed very small deep below the sediment-water interface but they are never really equal to zero (e.g. Bradley et al., 2020 https://doi.org/10.1126/sciadv.aba0697).

We have rephrased this sentence in the revised manuscript.

Eq. (7): What are FeSpro and H2Soxid? Are they different from the FeS and the H2S produced by the reactions shown in Eq. (5)?

We have corrected this inconsistency. The FeSprod and H2Soxid are names of the reaction terms as used in the code.

Why is one rate a capital R and the other a lower case r?

Text formatting issue with Microsoft when we translated our text from latex to Word. Corrected in the revised manuscript.

Eq. (9): What is the value of kads and can you give some information on this aspect of the model?

Kads is dimensionless quantity. In the revised we shall add:

"…*With special consideration of ammonium adsorption to sediment particles, the governing equation is given by:*

$$\frac{\partial \phi C}{\partial t} = -\frac{\partial}{\partial z}\left[-\frac{\phi \times D_{sed}}{(1+k_{ads})} \times \frac{\partial C}{\partial z} + w_\infty \times \phi_\infty \times C\right] + \Sigma \frac{\phi \times REAC}{(1+k_{ads})} \qquad (9)$$

*where we assumed that the immobilization of $NH_4^+$ is in instantaneous, local equilibrium (i.e. Any changes caused by the slow $NH_4^+$ removal process results in an immediate adjustment of the $NH_4^+$ equilibrium; so, can be modelled with a simple chemical species) and $k_{ads}$ is the adsorption coefficient. The inclusion of this formulation for the diffusion and reaction term has the effect of slowing down ammonium migration in sediment. Derivation of this formulation is given in Berner, 1980; Soetaert & Herman, 2009…*"

Kads has a value of 1.3 (Soetaert et al., 1996a) and is now include in the table of the supplementary text.

L216-220: How is irrigation implemented into the model, i.e., where does it appear in Eqs. (8 & 9)?

In the revised text, more detailed have been added. "Bio"-irrigation is implemented as a non-local transport term and contained in the REAC term in Eqs. 8 & 9.

L236: What is a "time run"?

Time is the duration of the model simulation. In the revised manuscript, this "time run" phrase has been clarified.

Eq. (15) Please precise here that TOCold is the TOC concentration at the old sedimentwater interface

This clarification has been added in the revised manuscript.

L248: it would be good to have more information on confac (alpha): here it is tuned. How should it be used in future applications? Always to the same value? Does its value depends on type and magnitude of flood?

See previous comment above. The concfac is tunable and we commented briefly on this parameter in section 4.2 L646 – "*This difference in characteristics was partially imposed in our study by assuming variations of α in the new deposit. The empirical observation of sediment characteristics associated with flood input dictates this parametric turning to match the TOC characteristics. ...However, more data from the field and laboratory experiments that resolve the OM composition of flood deposits are required to constrain the choice of this numerical parameter.*"

The parameter has been created to simulate organic enrichment depending on the magnitude and type of flood. It can be optimized to fit the shape and distinct characteristics of the TOC profile.

Fig.2: change "reactive Corg" for the notation "C (superscript)fast (subscript)org" for consistency

Label was changed in the Figure accordingly.

Table 2: the value for rslow is 0.0 d-1, but in the text it is indicated as 0.0031 d-1. Please clarify that

It was truncated during the alignment of the table to fit the page width. This was fixed in the revised manuscript.

Section 2.2.7: what about bottom boundary conditions? Is the concentration really set to 0 for all species, as indicated in Table S1? That would seem unjustified.

We have removed the useless dw parameters from table since all the bottom boundary condition are same and given as zero flux boundary.

L289: By sedimentation rate do you mean solid burial velocity? Porewater burial velocity? Both should be different because porosity is not constant with depth.

As sediment compaction is not included in the model, the advection rate for both solid and solutes are the same. In addition, for solutes, the effect of diffusion is order magnitude greater than transport due to porewater burial. This distinction has been made clear in the revised paper.

L289: Is w 0.027 or 0.03 cm per day? Be consistent between the text and tables.

The value was automatically rounded when fitting the table to page. We have updated the value in the text and table.

L283: Why a different NC ratio for both organic matter fractions? How were the values of 0.14 and 0.1 obtained? Why are these values different from those shown in Table 2?

Table 2 have been corrected. The two NC ratio were derived from Ait-Ballagh et al., 2020 optimized fit to data in the Rhône prodelta.

L306: Can you provide details (i.e., show the formula) on how are equations 8-10 integrated?

A new section will be added to that effect.

A snippet in this subsection is as follow:

*"Because the procedure is based on OMEXDIA, complete details of the derivation can be found in that paper and referenced therein (Soetaert et al., 1996). Here we recap the mathematical formulation of the method-of-lines (MOL) algorithm used by FESDIA. Direct differencing of (8) - (10) results to:*

$$\frac{\partial C_i}{\partial t} = \frac{\Phi_{i,i+1} D_{\Phi_{i,i+1}} (C_{i+1} - C_i)}{\Phi_i \Delta x_{i,i+1} \Delta x_i} - w_\infty \Phi_\infty \frac{\alpha_{i,i+1} C_i + (1 - \alpha_{i,i+1}) C_{i+1}}{\Phi_i \Delta x_i} - \frac{\Phi_{i-1,i} D_{\Phi_{i-1,i}} (C_i - C_{i-1})}{\Phi_i \Delta x_{i-1,i} \Delta x_i} + w_\infty \Phi_\infty \frac{\alpha_{i-1,i} C_{i-1} + (1 - \alpha_{i-1,i}) C_i}{\Phi_i \Delta x_i} \quad (22)$$

*for a generic tracer $C$ with a phase properties index $\Phi$ and $D_\Phi$ denoting porosity and dispersive mixing term respectively for solid or liquid. This equation is calculated such that the variables and parameters are defined both at the centre of each layer $x_i$ and at the interface between layers $(x_{i,i+1}, x_{i,i+1})$. The position at the centre of the grid is then given as $x_i = \frac{x_{i-1,i} + x_{i+1,i}}{2}$. $\Delta x_i$ represents the thickness of the i-layer and $\Delta x_{i,i+1}$ is the distance between two consecutive grid layers. A Fiadeiro scheme (Fiadeiro and Veronis, 1977) based on the model's Peclet number (a dimensionless ratio expressing the relative*

*importance of advective over dispersive processes) is used to provide a weighted difference of the transport terms in order to reduce numerical dispersion.*

*Equations (8)-(10) implemented as Eq. (22) is integrated in time using an implicit solver, called lsodes, that is part of the ODEPACK solvers (Hindmarsh, 1983). This solver uses a backward differentiation method (BDF); it has an adaptive time step, and is designed for solving systems of ordinary differential equations where the Jacobian matrix has an arbitrary sparse structure.*"

L306: Please provide guidance on what dt values should users set depending on the simulation

The solver internal timestep dt value is adaptive as explained above to ensure stability, consistency of solution and overcome model stiffness (Soetaert et al., 2010; Petzold 1983). The user dt for model is set based on the time resolution for which the processes of interest are needed. We have updated the manuscript to reflect this suggestion for more guidance.

L313: First time the "mix" perturbation is mentioned. What is that?

It has been removed.

Eq.(22) Why is it summed over the total number of grid points? Any perturbation following a flood should be the highest near the sediment-water interface, so wouldn't using data coming from deeper in the sediment to compute the relaxation time dilute the true signal and induce additional uncertainties?

The reviewer made an excellent point about the summation over the grid layer, which corresponds to some of our internal discussions while developing this metric. However, decision was made based on the following factors.

The perturbation signal can indeed be strongest near the sediment-water interface. This is true, however, only if the perturbation is small in size (thickness). Given the high variability of flood deposition seen from: Deflandre et al., 2002, Tesi et al 2012, Pastor et al., 2018 - such a "close-to-surface" assumption is not rarely observed. Furthermore, the perturbation signature spreads to deeper layer as the solute reprofile begins to reorganize. This reorganization is frequently nonlinear, and integrating over a limited domain may result in an incorrect relaxation timescale estimate, particularly when large perturbations occur, as shown in Figure 8. We argue in section 4.3 that this large depth interval related to their timescale for some species. For all these reasons, the whole domain it included in the calculation.

**Results**

Fig4: what is the alpha value for the slow organic carbon fraction?

The $\alpha$ value chosen in this scenario was 10. This information has been included in the figure caption.

L409-412: Can the authors interpret the mismatch between modelled and observed SO4, DIC and NH4 values at depth? Wouldn't that argue for overestimated organic carbon reactivities at depth?

The figure in submitted was fitted using parameter from Pastor et al., 2011, Ait-Ballagh et al., 2021 where advection rate is $w_\infty$= 0.03 cm/d. The figure represents an independent model validation without further calibration. With a slightly higher advection rate ($w_\infty$ = 0.05 cm/d), a better fit to the data can be reproduced. However, the final relaxation time using this modified advection rate stays within the uncertainty already shown in the manuscript. The revised manuscript was updated accordingly.

[Figure]

Table 3: How can there be an oxygen flux to the sediment that is ten times smaller than a DIC flux from the sediment? Wouldn't a value closer to one be expected?

The $O_2$ and DIC fluxes are different as supported by measurement from Rassman et al 2020, where DIC fluxes are 8 times higher than dissolved oxygen uptake (DOU). In the proximal station considered here, anaerobic pathways are dominant in comparison to the aerobic one (Pastor et al 2011). Hence, the DIC flux is expected to be significantly higher than the $O_2$ flux and not expected to be follow a 1:1 relationship.

Fig.9: what is "degradable OM"? does that mean that the alpha value is the same for both fast- and slow-decay organic carbon? If so, precise it.

It should be notable that both fraction of OM is degradable here. Only their timescale of degradation change. The $\alpha$ value was chosen to span the spectrum of what might be consider rich OC deposit and poor OC deposit.

L524: "mixing events" are again mentioned as something the model is able to simulate, but they are not described earlier, so it is unclear what they are.

The "mixing event" has been removed. This has been corrected to "deposition events"

Section 4.4: to add to this discussion, and in reference to the mention earlier in the manuscript of a "perturbed trajectory frequently arbitrarily divided into a fast, transient phase and a slow, asymptotic stage": should we instead think about relaxation time as the time necessary for most of, rather than all, changes to occur, similar to the concept of half-life in radioactivity?

That is another perspective we can consider as related to the relaxation timescale. However, it should be noted that this is predicated on the assumption that the curve is exactly exponential, which is not always the case in the simulated data we computed. With an analytical exponential fit, we can capture the main trend of the curve but it is difficult to estimate the time point when the event disappears.

Reference [Ait Ballagh et al., 2021] is missing from the list

This has been updated in the revised manuscript.

**REFERENCES**

Alan C. Hindmarsh, ODEPACK, A Systematized Collection of ODE Solvers, in Scientific Computing, R. S. Stepleman et al. (Eds.), North-Holland, Amsterdam, pp. 55-64, 1983.

Ait-Ballagh, F. E., Rabouille, C., Andrieux-Loyer, F., Soetaert, K., Elkalay, K., and Khalil, K.: Spatio-temporal dynamics of sedimentary phosphorus along two temperate eutrophic estuaries: A data-modelling approach, Continental Shelf Research, 193, 104037, publisher: Elsevier, 2020.

Boudreau, B. P.: Is burial velocity a master parameter for bioturbation?, 58, 1243–1249, 1994.

Boudreau, B. P.: A method-of-lines code for carbon and nutrient diagenesis in aquatic sediments, Comput. Geosci., 22, 479–496, https://doi.org/10.1016/0098-3004(95)00115-8, 1996b.

Boudreau, B. P.: Diagenetic models and their implementation, vol. 410, Springer, Berlin, 1997.

Burdige, D. J.: Burial of terrestrial organic matter in marine sediments: A re-assessment, Global Biogeochemical Cycles, 19, 2005. 690

De Borger, E., Tiano, J., Braeckman, U., Rijnsdorp, A. D., and Soetaert, K.: Impact of bottom trawling on sediment biogeochemistry: a modelling approach, Biogeosciences, 18, 2539–2557, 2021.

Deflandre, B., Mucci, A., Gagné, J.-P., Guignard, C., and jørn Sundby, B.: Early diagenetic processes in coastal marine sediments disturbed by a catastrophic sedimentation event, Geochimica et Cosmochimica Acta, 66, 2547–2558, 2002.

Hofmann, A. F., Meysman, F. J. R., Soetaert, K., and Middelburg, J. J.: A step-by-step procedure for pH model construction in aquatic systems, Biogeosciences, 5, 227–251, doi:10.5194/bg-5- 227-2008, 2008.

Boudreau, B. P.,and B. B. Jørgensen. The Benthic Boundary Layer: Transport Processes and Biogeochemistry, Oxford Univ. Press, Oxford, N. Y, 2001.

Katsev, S., Sundby, B., and Mucci, A.: Modeling vertical excursions of the redox boundary in sediments: Application to deep basins of the Arctic Ocean, Limnology and Oceanography, 51, 1581–1593, https://doi.org/10.4319/lo.2006.51.4.1581, 2006.

McKee, B. A., Aller, R., Allison, M., Bianchi, T., and Kineke, G.: Transport and transformation of dissolved and particulate materials on continental margins influenced by major rivers: benthic boundary layer and seabed processes, Continental Shelf Research, 24, 899–926, 2004.

Munhoven, G.: Model of Early Diagenesis in the Upper Sediment with Adaptable complexity – MEDUSA (v. 2): a time-dependent biogeochemical sediment module for Earth system models, process analysis and teaching, Geosci. Model Dev., 14, 3603–3631, https://doi.org/10.5194/gmd-14-3603-2021, 2021.

Pastor, L., Rabouille, C., Metzger, E., Thibault Chanvalon, A., Viollier, E.,and Deflandre, B.:Transient early diagenetic processes in Rhône prodelta sediments revealed in contrasting flood events, Continental Shelf Research, 166, 65–76, https://doi.org/10.1016/j.csr.2018.07.005, 2018.

Petzold, L. R. Automatic selection of methods for solving stiff and nonstiff systems of ordinary differential equations. SIAM Journal on Scientific and Statistical Computing, 4, 136–148, 1983.

Rassmann, J., Eitel, E. M., Lansard, B., Cathalot, C., Brandily, C., Taillefert, M., and Rabouille, C.: Benthic alkalinity and dissolved inorganic carbon fluxes in the Rhône River prodelta generated by decoupled aerobic and anaerobic processes, Biogeosciences, 17, 13–33, https://doi.org/10.5194/bg-17-13-2020, 2020.

Rassmann, J., Lansard, B., Pozzato, L., and Rabouille, C.: Carbonate chemistry in sediment porewaters of the Rhône River delta driven by early diagenesis (northwestern Mediterranean), Biogeosciences, 13, 5379–5394, 2016.

Sciberras, M., Hiddink, J. G., Jennings, S., Szostek, C. L., Hughes, K. M., Kneafsey, B., Clarke, L. J., Ellis, N., Rijnsdorp, A. D., McConnaughey, R. A., et al.: Response of benthic fauna to experimental bottom fishing: A global meta-analysis, 19, 698–715, 2018.

Soetaert, K., Herman, P. M., and Middelburg, J. J.: Dynamic response of deep-sea sediments to seasonal variations: A model, Limnology and Oceanography, 41, 1651–1668, 1996.

Soetaert, K., Herman, P. M., and Middelburg, J. J.: A model of early diagenetic processes from the shelf to abyssal depths, Geochim. Cosmochim. Acta, 60, 1019–1040, 1996.

Soetaert, K. and Meysman, F.: Reactive transport in aquatic ecosystems: Rapid model prototyping in the open source software R, Environmental Modelling & Software, 32, 49–60, 2012.

Soetaert, K., & Herman, P. A Practical Guide to Ecological Modelling: Using R as a Simulation Platform. Dordrecht: Springer, 2009. [doi:10.1007/978-1-4020-8624-3]

Soetaert, K., Petzoldt, T., and Meysman, F.: Marelac: Tools for aquatic sciences, 2010a.

Soetaert, K., Petzoldt, T., and Setzer, R. W.: Solving Differential Equations in R: Package deSolve, Journal of Statistical Software, 33, 1–25, https://doi.org/10.18637/jss.v033.i09, 2010b.

Soetaert, K.: Package rootSolve: roots, gradients and steady-states in R, Google Scholar, 2014.

Sulpis, O., Boudreau, B. P., Mucci, A., Jenkins, C., Trossman, D. S., Arbic, B. K., and Key, R. M.: Current CaCO3 dissolution at the seafloor caused by anthropogenic CO2, 115, 11700–11705, 2018.

Sulpis, O., Humphreys, M., Wilhelmus, M., Carroll, D., Berelson, W., Menemenlis, D., & Adkins, J. RADIv1: A non-steady-state early diagenetic model for ocean sediments in Julia and MATLAB/GNU Octave. Geoscientific Model Development, 15, 2105–2131. 2021

Sundby, B.: Transient state diagenesis in continental margin muds, Marine chemistry, 102, 2–12, 2006.

Tesi, T., Langone, L., Goñi, M., Wheatcroft, R., Miserocchi, S., and Bertotti, L.: Early diagenesis of recently deposited organic matter: A 9-yr time-series study of a flood deposit, Geochimica et Cosmochimica Acta, 83, 19–36, 2012.

---

## Author Response (AR1)

**AUTHOR'S RESPONSE**

Dear Editor,

Please find below our listing to changes in response to the reviewers' comments. We refer to our author comments at https://doi.org/10.5194/gmd-2022-84-AC2 and https://doi.org/10.5194/gmd-2022-84-AC3 for detailed responses. We also provide a tracked-change version of the manuscript highlighting the point-by-point insertions and deletions in the text alongside a revised manuscript. We hope these changes will address the reviewer's comments.

Best regards,

Stanley Nmor, on the behalf of the authors.

**REVISIONS IN RESPONSE TO COMMENTS BY ANONYMOUS REFEREE #1**

(1) Model description

It would be helpful if the authors can provide a short overview of model development relative to the previous and already published works/models. Current manuscript referred to these works, but what is exactly new is kind of obscure. Also, brief summary of the capacity/features of previous models from which the current model has been developed will be useful, including programing language, governing equations, and algorithms utilized for numerical solutions (finite difference/volume/element method for equation differencing and Newton iteration if adopted for solution seeking etc.). The current manuscript provides some of above information, but some information is still missing. And it is not 100% clear which part is new to the current model and which part is not new.

- **We added a paragraph (section 1.0) to the introduction that briefly describes how FESDIA differs from its other published models. Furthermore, a new sub-section has been added to the materials and methods (section 2.2.9 in revised manuscript) detailing FESDIA's numeric solution methods.**

(2) Definition of relaxation time

Eq. 22 describes the change rate of species concentration within sediment profile but does not necessarily define deviation from the pre-flood profile. Also, current manuscript lacks figures that directly compare the solute profile development relative to the pre-event profile. I think Figs. 4-8 need to be improved so that profile development relative to the pre-event profile is more visually obvious.

- **We explained why the profiles shown in Figures 4–8 were chosen because our paper only attempted to provide an estimate of when the perturbation disappears and not necessarily "return to the old profile" (see section 2.2.10 and section 4.5). The former concept is easily**

**quantifiable, whereas the latter may be difficult to estimate for a complex dynamical system such as those found in coastal sediment.**

(3) Description of model limitations and future development

Relevant to the above point, but limitations of model should be discussed more. There are several simplifying assumptions in the model but its influences on e.g., model validation, comparison with observation and estimation for relaxation time are not discussed. For instance, the authors assume that burial rate/porosity does not change with the flooding, but it is not discussed whether this assumption is defendable or close to what we observe. Model validation or comparison with the observed data is essentially based on solute profiles, which likely resulted from a good fit of TOC and may be achievable under different assumptions (those that allow changes of burial rate, porosity, bio-mixing and irrigation etc.). If this is the case, the relaxation time is likely quite different under different assumptions. While most of manuscript discussed how relaxation time is calculated under the specific assumption adopted for this study, it is not discussed how the relaxation time is affected by adopted assumptions.

- **We have added a new section on model limitations, as well as where future development will be made to FESDIA (see section 4.4 in revised manuscript). There we discuss possible biogeochemical processes which might affect the relaxation estimate we calculated in line with the reviewer's comment.**

**SPECIFIC COMMENTS**

L134. Three OM fractions? Thought the authors are using two.

- **In the revised paper, this sentence has been clarified by emphasizing that FESDIA only considers TWO reactive modelled fractions with the inert fraction parameterized using an asymptotic refractory carbon as background to diagnose the total organic carbon (TOC) from the simulation.**

L138. Froelich et al. (1979) rather than Froelich (1988) according to Bethke et al. (2011, AJS 311, 183)?

- **We apologize for this. This have been corrected in the revised manuscript (see L150 in the revision).**

L166. What does 'a coupled reaction formulation' mean?

- **This statement has been rewritten in L179 in the revised manuscript.**

Eqs. 6, 7. It does not make any sense to use/define 'maximum rates' when one is not using Monod or Michaelis-Menten type of equation.

- **We have corrected for the text by correctly removing the "maximum" from the sentence.**

Eq. 7. What is the definition of rH2Soxid?

- **This is have removed as it was duplicated in the original text.**

Eqs. 8, 9. What is the assumption behind the formulation of these equations? For instance, how do you obtain Eq. 9 for aqueous $NH_4^+$ with accounting for adsorbed $NH_4^+$? Can you track $NH_4^+$ adsorbed onto solid species along with OM and other solids or do you have to simplify that adsorption is depthindependent and/or time-independent? This can be important if solid materials with unoccupied exchange sites are flooded to sediment depocenter in a short period of time. If such case is possible, one would expect a large sink of $NH_4^+$ to the exchange sites? Related to this, do you model $PO_4$ adsorption onto Fe hydroxides or you do not have to do this?

- **The assumption underlying this formulation has been clarified in section 2.2.3 L195-199 of the revised manuscript. We did not discuss the effect of $PO_4$ because the described model does not include it. However, we explain how adsorption affects NH4 diffusion.**

L205. Does porosity 'decay'?

- **We have changed the phrase according in revised manuscript in L224.**

L209. According to Eq. 10, the authors seem to assume intraphase biodiffusion (Meysman et al., 2005, GCA 69, 3601). The statement here mentioning an interphase biodiffusion is inconsistent with Eq. 10.

- **This mistake as used in this particular sentence have been corrected to "*interphase*" to reflect how bioturbation is modelled in FESDIA (in L228 of revised manuscript).**

Eq. 14. Irrigation term is not found in the governing equation. Is it included as a reaction term?

- **We have stated clearly where that the REAC term in Eq 8 – 9 includes the irrigation term in L191-194.**

Section 2.2.5 & Section 2.2.6. More details are desirable as adding grid for implementing a deposition event must be an important addition to the previous modeling framework. For instance, how you define Zpert, e.g., number of grids and their geometry etc. Some examples, not only schematics may also be useful.

- **We have explained how the vertical grid was created using other community tools for early diagenesis. There is a reference to those tools. The grid layer schematic is shown in the supplementary figure (Fig S1).**

L276. 'maximum in the spring and minimum in fall and winter'. This line does not make sense to me.

- **This ambiguity in our choice of word have been corrected in the reviewed manuscript (in L305 for revised manuscript).**

Eq. 21. What is the units of TOC? Also, how do you derive Eq. 21? Is this simply analytical solution of the governing equation? In any case, it would be helpful if the authors can provide the procedure to obtain Eq. 21 somewhere.

- **This part of section 2.2.8 have been rewritten to entail to the TOC calculation, unit and how it was derived.**

L321. 'were utilizes the R programming language'. Correct English?

- **We apologize for this grammatical mistake. It has been corrected in the revised manuscript.**

Section 2.2.9. It would be helpful if the authors can make a short description of what part of "deSolve package" they used, not only directing the reader to the R-forge webpage. More specifically, how the authors numerically solve the governing equations, apart from "method-of-lines" methods? Use of any finite difference/volume/element method?

- **We have added a new subsection (see section 2.2.9) to reflect this remark and the method used to solve the PDE (in L333 of revised manuscript).**

How is the time-integration of governing equations made (time-implicitly or -explicitly)?

- **New information on this is added in section 2.2.9 of the revised manuscript.**

L331. What is a "slow" stationary state?

- **We have clarified to the remark of the phrase above. This sentence has been paraphrased better (in L372).**

Eq. 22. Not quite sure this is a legit mathematical expression. phi(tau) < threshold is what I thought is consistent with what the authors described.

- **To aid comprehension and consistency with the textual explanation, we modified this equation block to explicitly include the threshold subscript (in L377 of the revised manuscript).**

L334. "threshold (i.e given by the median over the entire time duration)." Do you mean that the run is finished when difference becomes less than the median value throughout the simulation and then tau is defined as the model time required for this?

- **No change required. We provided answer our affirmative answer to the reviewer.**

L340. I probably do not fully understand the ensemble of simulations here to estimate the uncertainty in tau. What parameter do you randomly re-sampled exactly? Median of the reference run through time? If so, the runs for determining the uncertainty in tau is conducted until (randomly-chosen) prescribed median is crossed? But this does not necessitate re-running of the model as the boundary conditions are not changed?

- **This section 2.2.10 has been rewritten to better convey the nature of the ensemble run calculation for determining the uncertainty.**

L383. "a thickness scale of 1 cm to 30 cm in 5 cm increments". This line does not make sense to me. What exactly did you use for thickness in sensitivity analysis?

- **This line has been rewritten (in L434 in the revised manuscript).**

L507. Please specify what "RiOmar" stands for.

- **This is a common abbreviation used by researchers in this field. We expanded on this definition in the introduction, where it was first mentioned.**

L621. above → below?

- **This word and its surrounding sentence have been changed (in L674 of the revised manuscript).**

--- TECHNICAL COMMENTS ---

- **The minor fix for the technical comments listed by the reviewer have been corrected accordingly in the revised manuscript.**

**REVISIONS IN RESPONSE TO COMMENTS BY ANONYMOUS REFEREE #2**

**GENERAL COMMENT**

Rassmann et al. (2016 https://doi.org/10.5194/bg-13-5379-2016, 2020 https://doi.org/10.5194/bg-17-13-2020) described Rhône river delta sediments rich in calcium carbonates, and reported signs of the various reactions associated with calcium carbonates occurring in those sediments. If the focus here is Rhône river delta sediments, why not including any calcium carbonate species in the current model?

- **In our response to the reviewer's comment we have address reasons why modelling the carbonate system wasn't modelled by FESDIA. We re-emphasis the primary goal of work presented here is to design a model to study the first order effects of flood deposition on the profiles (we argue that the effect of calcite formation is contribute <15% to the overall**

**variation in DIC profile which could help justify our choice), as well as to highlight the possibility of incorporating diagenesis changes in the sediment when including abrupt discontinuities. Given this computational demanding nature of incorporating carbonate chemistry as reflected in other model, we have omitted including such dynamics in the first working version of FESDIA. Effort are underway to bring this interesting carbonate calculation possibility to FESDIA. To that effect, we discuss this as some model limitation in section of 4.4 of the revised manuscript.**

Section 2.2.6: this seems to be the most important section in terms of model development, but it is also the part I had the hardest time to follow. There are a lot of new terms introduced here and they are not well defined. The second paragraph starts to explain how post-flood organic carbon contents are derived, then mentions the solutes, then goes back to post-flood organic carbon content with Eq.(15)... I suggest putting more effort clarifying this section, defining terms with precise and consistent words, in order not to confuse the reader.

- **This section has been rewritten to improve how FESDIA handled this depositional event calculation. A new table (see Table 1 in revised manuscript) has been added to the revised manuscript to assist in providing summary context to some of the symbols used in the FESDIA.**

Specifically, what is the carbon enrichment factor (confac) exactly, and how does it differ from the proportionality constant (pfast)?

- **We have clarified where the differences between in the concfac and pfast exist in L280 of section 2.2.6 in the revised manuscript. Briefly, "...*This OC flux partitioning by $pfast$ occurs regardless of the event and it is related to the carbon flux received at the boundary, but the carbon enrichment factor occurs only during the event. The Carbon enrichment factor ($\alpha$) can be viewed as a method of imposing a new initial condition only at the time of the event ...*".**

What is $Corg\ flood$ and how does it differ from TOC (both are present in Eq. (15))? Is confac tuned for each simulation or is it constant? Is pfast tuned for each simulation or is it constant?

- **We respond to the reviewer's observation. We included a glossary of the symbols used in the manuscript in the revised manuscript (see Table 1). Furthermore, the original manuscript states that only We respond to the reviewer's observation. We included a glossary of the symbols used in the manuscript in the revised manuscript (see Table 1). Furthermore, the original manuscript states that only alpha was changed in the sensitivity runs, with all other parameters remaining constant. This is added in section 2.2.11.1, in L427.**

It is stated L187-188 that "For dynamic simulation, w can change as a function of time but in most cases, we assumed a constant value." In which cases exactly was w changing? Changing w in all cases seem like a necessity given that the novelty of the model is to simulate events in which the flux of deposited material (thus w) is strongly changing with time. How can a constant w be appropriate to simulate a flood? w also changes with sediment depth, because of chemical reactions occurring within the sediment (see Munhoven, 2021 https://doi.org/10.5194/gmd-14-3603-2021). Can the authors either better justify their choice of a non-changing w or update that in the model simulations?

- **We have responded to the reviewer's comment. To summarize, we argue that because episodic flood deposition (~10-30cm/d of deposition in about 5 days depending on the extremity of the event) dominates the average background variation in sedimentation rate in the Rhône prodelta (where our benchmark was conducted) and this instantaneous deposited is treated separately in FESDIA, we can justify that the main variation in sedimentation is already accounted for. Furthermore, FESDIA has the option of including time-dependent sedimentation rates if such data are available (see section 4.4). We also mentioned that the current version of FESDIA does not take into account depth dependency change of w due to chemical diagenesis, and we explained why in our response to the reviewer's comment.**

Section 2.2.7: in most O2 and pH microprofiles from the Rhône delta presented in Rassmann et al. (2016) we can see the influence of a diffusive boundary layer. Please discuss and justify the absence of diffusive boundary layer control on solutes as an upper boundary condition, or update the upper boundary condition accordingly to include this, as other models do in a simple manner (Boudreau et al., 1996 https://doi.org/10.1016/0098-3004(95)00115-8; Munhoven, 2021 https://doi.org/10.5194/gmd14-3603-2021; Sulpis et al., 2022 https://doi.org/10.5194/gmd-15-2105-2022)

- **This is a significant omission in the current version of FESDIA, and we addressed it in our response to the reviewer. We made some simplifications and assumptions in the boundary condition, which are now covered in section 4.4 of the revised manuscript. Essentially, our conclusion for the relaxation times calculation does not drastically change with the inclusion of the DBL as the deviation of our simple calculation suggests that the relaxation times were within the confidence interval shown in this paper. However, we acknowledge that if oxygen fluxes are to be estimated, FESDIA in its current form will overestimate oxygen fluxes but less so for solutes such as DIC and SO$_4$. FESDIA will almost certainly include diffusive boundary layer dynamics in future versions. Section 4.5 of the revised article contains additional explanation (see L715).**

**SPECIFIC COMMENTS:**

Shouldn't "Rhône" be spelled "Rhône", even in English language?

- **The proper French spelling have been corrected in all instance within the manuscript.**

L35: The use of the acronym RiOmar is not really needed, since only used once after. In general, avoid unnecessary acronyms.

L36: Although more commonly used, POC is also an unnecessary acronym here, since only used once after.

- **We have corrected the text and removed redundant acronyms that were not used elsewhere in the manuscript.**

L36-39: The sentence is unclear. "because it serves as a sink for particulate organic carbon and nutrients as well as an intense site of carbon and nutrient": what is the "it" referring to?

- **The sentence has been rewritten to improve clarity (see L39 in the revised manuscript).**

L40: I am not convinced that all the cited models have time-dependent capabilities, unlike several other, more recent models, published in this journal that explicitly do. Please update the list.

- **An updated list of models has been added (see L42 in the revised manuscript).**

L43: "massive episodic events" could refer to lots of processes, please be more specific

- **The sentence has been modified (see L44).**

L47-50: Sentence unclear. "Attempts to use mathematical models to understand perturbation-induced events on early diagenetic processes have resulted in a variety of approaches that incorporate this type of local phenomenon.": what is "this type of local phenomenon" referring to?

- **The sentence has been changed to refer to the "phenomenon" (see L49).**

L48-50: "As an example, previous research in deep-sea systems suggests that megafaunal perturbation can cause a 35% increase in silicic flux when compared to steady state estimates (Rabouille and Gaillard, 1990)" this is interesting but this level of precision seems unnecessary, what is the relevance for this study? Besides, what is a "silicic flux"? In which direction is the mentioned flux going?

- **This entire sentence has been deleted because its omission has no effect on the overall narrative of the paragraph block.**

L50: What is the "redox boundary"

L52-53: What does the "redistribution of solid-phase manganese with multiple peaks" mean?

- **The sentence has been changed to explain what a "redox boundary" is and why "redistribution of solid-phase manganese with multiple peaks" could occur (see L50-54 in revised manuscript).**

L62: "porewater species like oxygen (O2) can be restored after a few months": it is unclear. Do you mean that porewater concentrations can be restored to their pre-flood levels?

L66: what does "short-lived species" mean?

L67: DIC is a component, not a species.

- **We have made correction to the text according (see L62, L67 -68).**

Materials and methods

L93: "the organic matter delivered reflects the Rhône River inputs (Lansard et al., 2008; Cathalot et al., 2013)", in terms of what? Composition? Reactivity? Fig.1: I assume that the dashed and solid gray linings shown on the map depict bathymetry; it would be useful to precise it in a caption/legend.

- **The sentence "organic matter delivered..." has been changed to reflect the origin and compositional nature. The map now has a caption (see L101 in the revised manuscript).**

L107: what does "mode of behavior" mean?

L140: how exactly do "the reactivities decrease with depth" in the present model? From Table S1, it seems that the reactivities are constant.

L140: The sentence formulation is awkward: it is the degradation that would "cease", not its rate. Saying this also slightly exaggerated, degradation rates become indeed very small deep below the sedimentwater interface but they are never really equal to zero (e.g. Bradley et al., 2020 https://doi.org/10.1126/sciadv.aba0697).

Eq. (7): What are FeSpro and H2Soxid? Are they different from the FeS and the H2S produced by the reactions shown in Eq. (5)?

Why is one rate a capital R and the other a lower case r?

- **In the revised manuscript, all points have been noted and changed. The word choice and syntax errors have been corrected.**

Eq. (9): What is the value of kads and can you give some information on this aspect of the model?

- **This information is provided in 2.2.3, and the kads value of 1.3 has been added to the main text and parameter table (see Table 3 and L197).**

L216-220: How is irrigation implemented into the model, i.e., where does it appear in Eqs. (8 & 9)?

- **We addressed this issue by stating that irrigation is included in Equations 8 and 9, and a comment on this has been added to the revised manuscript (see L205 and L235).**

L236: What is a "time run"?

- **This has been rewritten to be more understandable (see L255).**

Eq. (15) Please precise here that TOCold is the TOC concentration at the old sedimentwater interface

- **This distinction has been added to the text, and a table with a summary of the notation used has been provided (see Table 1 and L268).**

L248: it would be good to have more information on confac (alpha): here it is tuned. How should it be used in future applications? Always to the same value? Does its value depends on type and magnitude of flood?

- **This comment on the confac's flexibility can be found in sections 2.2.6 and 4.2 of the original text.**

Fig.2: change "reactive Corg" for the notation "C (superscript)fast (subscript)org" for consistency

Table 2: the value for rslow is 0.0 d-1, but in the text it is indicated as 0.0031 d-1. Please clarify that

- **Both have bee corrected and updated in the figure and table respectively (see Table 3).**

Section 2.2.7: what about bottom boundary conditions? Is the concentration really set to 0 for all species, as indicated in Table S1? That would seem unjustified.

- **This has been removed from the parameters list because all species have the same "zero flux boundary."**

L289: By sedimentation rate do you mean solid burial velocity? Porewater burial velocity? Both should be different because porosity is not constant with depth.

- **In our response to the reviewer, we addressed this question. FESDIA assumed that the advection rate is the same for both solid and solute, but that the diffusion effect is more dominant as transport for porewater species than for solute advection.**

L289: Is w 0.027 or 0.03 cm per day? Be consistent between the text and tables.

- **We have changed that and made both entry consistent (see L317 and Table 3).**

L283: Why a different NC ratio for both organic matter fractions? How were the values of 0.14 and 0.1 obtained? Why are these values different from those shown in Table 2?

- **In our response to the reviewer, we explain why the NC ratio for each fraction was different. The table value has been corrected.**

L306: Can you provide details (i.e., show the formula) on how are equations 8-10 integrated?

L306: Please provide guidance on what dt values should users set depending on the simulation

- **We provided this information in section 2.2.9 of the revised manuscript.**

L313: First time the "mix" perturbation is mentioned. What is that?

- **The "mix" phrase has been corrected.**

Eq.(22) Why is it summed over the total number of grid points? Any perturbation following a flood should be the highest near the sediment-water interface, so wouldn't using data coming from deeper in the sediment to compute the relaxation time dilute the true signal and induce additional uncertainties?

- **We provided our justification of why the whole grid was used for the calculation of the relaxation timescale in our response to the reviewer's comment.**

**Results**

Fig4: what is the alpha value for the slow organic carbon fraction?

- **The value of alpha has been included and updated in the Figure 4.**

L409-412: Can the authors interpret the mismatch between modelled and observed SO4, DIC and NH4 values at depth? Wouldn't that argue for overestimated organic carbon reactivities at depth?

- **We have provided a remark on the misfit, made an adjustment to a parameter (sedimentation rate - w) and updated the result accordingly.**

Table 3: How can there be an oxygen flux to the sediment that is ten times smaller than a DIC flux from the sediment? Wouldn't a value closer to one be expected?

- **No changes were made here. We have provided a response to this in our reply to the reviewer.**

Fig.9: what is "degradable OM"? does that mean that the alpha value is the same for both fast- and slowdecay organic carbon? If so, precise it.

- **This has been clarified in the revised manuscript with comment on the alpha value used have been mentioned previously (see L545).**

L524: "mixing events" are again mentioned as something the model is able to simulate, but they are not described earlier, so it is unclear what they are.

- **"These mixing events" have been removed and the text updated.**

Section 4.4: to add to this discussion, and in reference to the mention earlier in the manuscript of a "perturbed trajectory frequently arbitrarily divided into a fast, transient phase and a slow, asymptotic

stage": should we instead think about relaxation time as the time necessary for most of, rather than all, changes to occur, similar to the concept of half-life in radioactivity?

- **We have responded to the half-life concept alluded to by the reviewer in our reply.**

Reference [Ait Ballagh et al., 2021] is missing from the list

- **The reference has been updated.**